# Dismantling Pathological Shortcuts: A Causal Framework for Faithful LVLM Decoding

Liu Yu [1 2]  Can Chen [1]  Ping Kuang [1]  Zhikun Feng [1]  Fan Zhou [1]  Gillian Dobbie [2]

## Abstract

Large Vision-Language Models (LVLMs) exhibit sophisticated reasoning but remain susceptible to object hallucination. Deviating from the prevailing *attention intensity assumption*, we reveal a deeper dynamic structural misalignment: hallucination is triggered at decision-critical steps where specific attention heads, acting as risky mediators, decouple from visual evidence to lock onto language priors. This establishes a pathological shortcut that bypasses visual grounding. To dismantle this, we propose **Fox** (Faithfulness and Observational-flow via eXpression-rectification), a training-free inference-time framework. Fox diagnoses structural misalignment using a visual attention entropy probe to localize risky mediators unsupervisedly. We then execute a targeted causal intervention via numerical logit saturation to physically sever the shortcut path. Finally, a conflict-gated cooperative decoding strategy reconciles interventional faithfulness with observational fluency. Extensive experiments demonstrate that Fox achieves SOTA performance, outperforming SID by 29.1% while preserving linguistic richness. Code is available at https://github.com/Cc2021start/Fox.

## 1 Introduction

Large Vision-Language Models (LVLMs) have demonstrated remarkable capabilities in multimodal reasoning (Liu et al., 2023; Wan et al., 2025). Despite these advancements, they frequently suffer from object hallucination—generating content that contradicts visual evidence (Leng et al., 2024a; Nie et al., 2025). This poses severe risks in safety-critical

[1]School of Information and Software Engineering, University of Electronic Science and Technology of China, Chengdu, Sichuan, China [2]School of Computer Science, University of Auckland, Auckland, New Zealand. Correspondence to: Ping Kuang <kuangping@uestc.edu.cn>.

*Proceedings of the 43rd International Conference on Machine Learning*, Seoul, South Korea. PMLR 306, 2026. Copyright 2026 by the author(s).

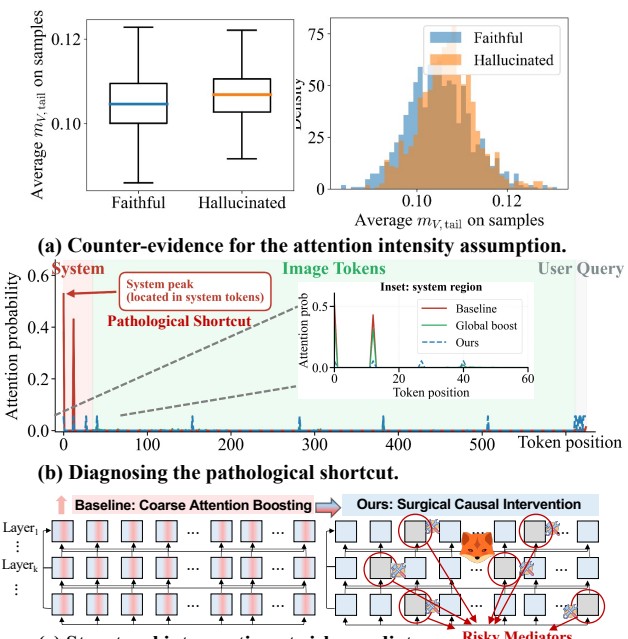

*Figure 1.* **Motivation of our work.** **(a)** Global visual attention magnitude $m_{V,tail}$ and distribution lack discriminative power to identify hallucination. **(b)** While global magnitude boosting (Green) fails to suppress the pathological peak on system instructions at decision-critical steps, our structural intervention (Blue) on *risky mediators* eliminates this shortcut, restoring visual grounding. **(c)** Unlike coarse-grained enhancement across all layers (Left), Fox performs a sparse, surgical intervention on diagnosed risky mediators (Right), physically severing the prior-driven shortcut.

domains, such as medical imaging or embodied AI (Wang et al., 2023; Tian et al., 2024), where a single hallucinated token can trigger catastrophic reasoning failures.

Current mitigation strategies generally fall into training-time alignment (Bai et al., 2025; Liu et al., 2024a) or inference-time intervention (Leng et al., 2024b; An et al., 2025; Li et al., 2025; Fazli et al., 2025). While training-based methods incur substantial computational costs, inference-time interventions have gained traction for their model-agnostic efficiency (Zhang et al., 2025; Chen et al., 2024; Che et al., 2025). Despite technical variations, most existing approaches share a common premise, which we term the *attention intensity assumption*: hallucination is primarily attributed to a quantitative deficit in visual atten-

tion (Chen et al., 2025). Consequently, these methods seek to rectify failures by mechanically amplifying visual signals (e.g., PAI (Liu et al., 2024b)) or suppressing language priors (Leng et al., 2024b). However, this intuition proves empirically incomplete, particularly for strategies predicated on global magnitude enhancement. As shown in Fig. 1(a), $m_{V,tail}$ denotes the visual attention magnitude, representing the total attention weight allocated to image tokens. A controlled analysis reveals no statistically significant reduction in global visual attention magnitude for hallucinated outputs, with their distributions largely overlapping. This lack of discriminative power suggests that focusing solely on intensity overlooks the underlying structural misalignment of hallucination. The decisive failure is therefore not only how much visual mass is assigned, but where the final prediction is routed at the moment of content generation. More details *cf.* Appendix A.1 and A.2.

Motivated by this, we shift our focus from global magnitude to the transient pathology triggered at decision-critical steps. We observe that hallucination is driven by specific attention heads, i.e., *risky mediators*, that functionally decouple from visual evidence precisely when the model commits to content-bearing generation. As depicted in Fig. 1(b), a naive global boosting strategy (e.g., PAI (Liu et al., 2024b)) succeeds in increasing the total attention volume but fails to dismantle the localized, pathological peak on system instructions. This persistent structural bias establishes a shortcut where latent language priors bypass visual grounding to dominate the output. From a causal perspective, these heads act as unreliable mediators that reroute influence via spurious dependencies. As shown in Fig. 1(c), addressing this requires a shift from uniform, token-level adjustments toward sparse, head-level causal interventions.

To dismantle this pathological structure, we propose Fox (Faithfulness and Observational-flow via eXpression-rectification), a training-free framework grounded in a Structural Causal Model (SCM). We reformulate decoding as a causal process where attention heads at specific decision-critical steps serve as mediators. Specifically, we introduce visual attention entropy as an unsupervised probe to pinpoint risky mediators exhibiting high visual uncertainty. Upon detection, we execute a targeted intervention via the do-operator—implemented as numerical logit saturation—to physically sever the shortcut path, forcing the model to rely on direct visual evidence. Finally, to reconcile interventional faithfulness with linguistic fluency, we implement a conflict-gated cooperative decoding strategy that dynamically fuses observational and interventional distributions. Our main contributions are summarized as follows:

- We challenge the prevailing attention intensity assumption by revealing that hallucination stems from dynamic structural misalignment. We identify risky mediators—sparse

heads that structurally disconnect from visual inputs at decision-critical steps—offering a novel mechanistic perspective on LVLM failures.
- We propose Fox, a principled inference-time framework rooted in SCM. By intersecting decision-critical steps with visual attention entropy probes, we achieve precise, unsupervised localization and do-driven suppression of pathological shortcuts.
- Extensive experiments demonstrate that Fox significantly outperforms existing baselines, achieving a 22.9% improvement on CHAIR and mitigating hallucination while preserving descriptive richness.

## 2 Related Work

**Hallucination Mitigation in LVLMs.** Existing strategies generally fall into training-time alignment (Sun et al., 2023; Zhou et al., 2024) and inference-time intervention (Zhu et al., 2026; Tong et al., 2025; Yu et al., 2026). Given the substantial cost of retraining, recent research has favored inference-time methods: contrastive decoding approaches such as VCD (Leng et al., 2024b) and ICD (Wang et al., 2024) penalize language priors via negative constraints, while reweighting methods (Liu et al., 2024b; Zou et al., 2024) mechanically amplify visual signals. More granular studies, such as OPERA (Huang et al., 2024b) and SEVI (Zhao et al., 2025), attempt to regulate generation by penalizing over-trust or emphasizing specific semantic layers. However, they predominantly hinge on the attention intensity assumption, treating the magnitude of attention as the primary proxy for faithfulness. Unlike these intensity-based heuristics, we argue that hallucination stems from a dynamic structural misalignment. By shifting the focus from global magnitude to the visual attention entropy at decision-critical steps, we distinguish valid reasoning from confident but misaligned hallucinations, offering a more precise diagnostic granularity.

**Causal Inference in Multimodal Reasoning.** Structural Causal Models (SCMs) provide a rigorous framework for debiasing and interpretability in vision-language tasks (Pearl, 2009). Related efforts use invariant learning, mixup, generated sentences, and graph contrastive pre-training to mitigate bias or enrich pretrained models (Zhou et al., 2023; Mao et al., 2023; Yu et al., 2023; 2024; 2025a;c), while bimodal debiasing extends this principle to text-to-image generation (Yu et al., 2025b). Recent studies like CausalMM (Zhou et al., 2025) and Huang et al. (2024a) employ SCMs to analyze hallucinations, typically utilizing input-level counterfactuals—such as masking image regions or tokens—to estimate causal effects. While effective for post-hoc diagnosis, these input-level perturbations are often too coarse to rectify the model's internal reasoning dynamics. In contrast, we reformulate internal attention heads as dynamic mediators. This allows us to perform

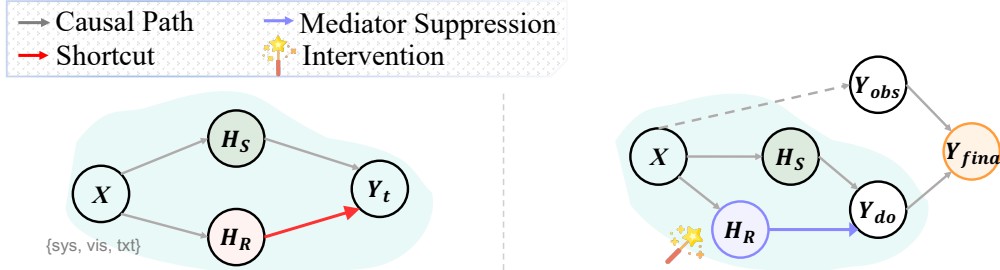

(a) Observation: risky heads dominate generation.  (b) Intervention: soft causal suppression of risky mediator.

*Figure 2.* **Structural Causal Model (SCM) of the LVLM Decoding Path.** (a) **Observational SCM:** The latent mediators $H$ are localized at decision-critical steps. While stable mediators $H_S$ maintain visual grounding, risky mediators $H_R$ trigger a **pathological shortcut** (red arrow) from language priors $\mathbf{X}_{sys}$ to output $Y_t$. (b) **Interventional SCM:** By applying $\mathbf{do}(H_R)$, we sever the shortcut. The final output $Y_{\text{final}}$ is dynamically reconciled from observational ($P_{obs}$) and interventional ($P_{do}$) distributions.

surgical interventions via the **do**-operator directly on the latent information flow when the model reaches decision-critical queries. By physically severing pathological shortcut paths within the network rather than altering external inputs, Fox achieves a principled, training-free restoration of visual grounding.

## 3 Preliminary

**Problem Formulation.** We consider an LVLM $\mathcal{F}_\theta$ that processes a multimodal input $\mathbf{X}$ partitioned into three semantic subspaces: visual $\mathbf{X}_{vis}$ (indices $\mathcal{I}_{vis}$), system instructions $\mathbf{X}_{sys}$ ($\mathcal{I}_{sys}$), and textual history $\mathbf{X}_{txt}$ ($\mathcal{I}_{txt}$). The model generates $Y = \{y_1, \ldots, y_L\}$ autoregressively, where the next-token probability is $P(y_t \mid \mathbf{X}, y_{<t}) = \text{Softmax}(\mathcal{F}_\theta(\mathbf{X}, y_{<t}))$. The internal routing mechanism is driven by multi-head attention. For head $h$ at layer $l$, the attention logits $\mathbf{L}^{(l,h)}$ and weights $\mathbf{A}^{(l,h)}$ are computed as:

$$\mathbf{L}^{(l,h)} = \frac{\mathbf{Q}^{(l,h)}(\mathbf{K}^{(l,h)})^\top}{\sqrt{d_k}}; \quad \mathbf{A}^{(l,h)} = \text{Softmax}(\mathbf{L}^{(l,h)}).$$

Our causal intervention (§ 4) directly modulates $\mathbf{L}^{(l,h)}$ to rectify the information flow before normalization.

**Causal Formulation.** We formalize the decoding process as an SCM in Fig. 2. We identify the attention heads acting at decision-critical steps $\mathcal{Q}$ as the dynamic mediators $H$, which transmit causal influence from inputs to output. We posit that causal mediation is temporally sparse, where information is aggregated at specific nodes ($q \in \mathcal{Q}, h \in \mathcal{H}$) rather than uniformly across tokens. Ideally, faithful generation requires $H$ to reliably transmit evidence via the visual path: $\mathbf{X}_{vis} \to H \to Y_t$.

However, we observe a structural misalignment of mediation in Fig. 2(a), where mediators bifurcate into: (1) Stable mediators $H_S$, maintaining grounded visual attention; (2) Risky mediators $H_R$, which functionally decouple from visual evidence to lock onto language priors. Specifically,

$H_R$ establishes a pathological shortcut: $\mathbf{X}_{sys} \to H_R \to Y_t$. While any text carries priors, $\mathbf{X}_{sys}$ (e.g., "You are a helpful assistant") serves as the primary anchor for latent priors when visual grounding fails, leading to object hallucination.

**Causal Intervention.** To block this shortcut without retraining, we apply the intervention $\mathbf{do}(H_R := \text{noise})$. This operation suppresses $H_R$, forcing the model to rely on the faithful visual path. This process yields two distributions: the observational anchor $P_{obs}$ and the interventional candidate $P_{do}$. As shown in Fig. 2(b), the final output $Y_{\text{final}}$ is derived via a conflict-gated causal fusion that dynamically reconciles these branches:

$$Y_{\text{final}} \sim \text{Softmax}(f_{\text{gate}}(\mathbf{z}_{obs}, \mathbf{z}_{do})). \tag{1}$$

The overall framework is guided by three research questions (RQs) targeting diagnosis, intervention, and cooperation. The Algorithm and detailed methodology *cf.* Appendix B.

## 4 Method

### 4.1 Causal Diagnosis of Risky Mediators

**RQ1.** *Given the dense multi-head attention mechanism in LVLMs, how can we unsupervisedly identify the specific attention heads that facilitate the pathological shortcut $\mathbf{X}_{sys} \to H_R \to Y_t$?*

*Insight I:* Hallucination is a dynamic structural misalignment rather than a uniform signal deficit. From a causal perspective, the breakdown of visual grounding is concentrated at specific nodes where the model aggregates multimodal context to update its internal states. Diagnosing hallucination thus requires dual-axis localization: identifying the intersection of decision-critical steps (temporal axis) and risky mediators (spatial axis).

**Temporal Axis: Identifying Decision-Critical Steps.** In autoregressive transformers, information flow is anisotropic

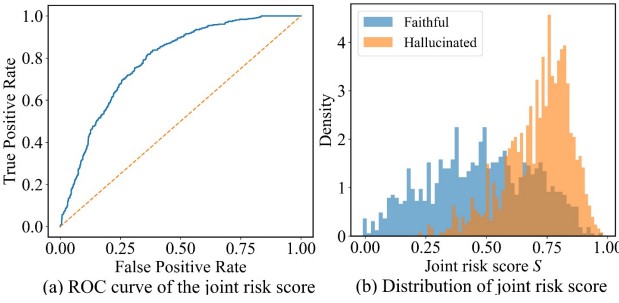

(a) ROC curve of the joint risk score     (b) Distribution of joint risk score

*Figure 3.* **Empirical validation of the joint risk score** ($S$). **(a) Diagnostic fidelity:** The ROC curve demonstrates that the aggregated joint risk score reliably distinguishes hallucinated trajectories from faithful ones (AUC=0.818). **(b) Structural decoupling:** The distribution shift confirms that hallucination (orange) is characterized by higher joint risk, signifying the concurrent collapse of visual reliability and activation of language priors. **Score construction:** The sample-level score is an aggregation of head-level joint risk $S^{(l,h)}$ weighted by their respective causal contributions (Eq. (4)).

and temporally sparse. We posit that the shortcut mechanism is most detectable at decision-critical steps $\mathcal{Q}$, where the model transitions from prompt encoding to token synthesis. Unlike prior methods that average signals across all tokens, we pinpoint two pivotal temporal anchors derived from the input $\mathbf{X}$ and history $y_{<t}$:

- **The Multimodal Handshake** ($\mathbf{x_{last}} \in \mathbf{X}$): The terminal token of the prefix sequence $\mathbf{X}$. As the final node of multimodal integration, $\mathbf{x}_{last}$ acts as the aggregator that compresses the visual-linguistic context ($\mathbf{X}_{vis}, \mathbf{X}_{sys}$) into the initial hidden state. A failure to ground on $\mathbf{X}_{vis}$ here leads to a corrupted trajectory initialization.
- **The Autoregressive Anchor** ($y_{t-1}$): The immediate predecessor of the current prediction $y_t$. As the proximal causal parent in the SCM, $y_{t-1}$ serves as the most sensitive probe for immediate prior dominance by language-biased heads during the generative phase.

**Spatial Axis: Quantifying Structural Shortcuts.** At the identified steps $\mathcal{Q}$, we inspect the latent mediators (attention heads) to detect structural decoupling. Following the subspaces defined in § 3, we partition the key space into system ($\mathcal{I}_{sys}$), visual ($\mathcal{I}_{vis}$), and text ($\mathcal{I}_{txt}$) indices. We quantify the prior-path activation for head $h$ at layer $l$ by computing the system attention magnitude at $\mathcal{Q}$:

$$m_{\text{sys}}^{(l,h)} = \frac{1}{|\mathcal{Q}|} \sum_{q \in \mathcal{Q}} \sum_{k \in \mathcal{I}_{sys}} \mathbf{A}_{q,k}^{(l,h)}, \qquad (2)$$

where $m_{\text{sys}}$ measures the reliance on $\mathbf{X}_{sys}$ relative to the current multimodal context. More details of $m_{\text{sys}}$ *cf.* Appendix A.3.

**Unsupervised Diagnosis via Visual Attention Entropy.** A shortcut is only verified when the reliance on priors occurs alongside the collapse of visual reliability. To measure the causal uncertainty of the visual pathway $\mathbf{X}_{vis} \to H$, we

re-normalize the attention weights strictly over the visual indices $\mathcal{I}_{vis}$ to obtain the local distribution $\hat{\mathbf{A}}_q^{(l,h)}$:

$$\hat{\mathbf{A}}_{q,j}^{(l,h)} = \frac{\mathbf{A}_{q,j}^{(l,h)}}{\sum_{k \in \mathcal{I}_{vis}} \mathbf{A}_{q,k}^{(l,h)}}, \quad j \in \mathcal{I}_{vis};$$
$$H_{\text{vis}}^{(l,h)} = \frac{1}{|\mathcal{Q}|} \sum_{q \in \mathcal{Q}} \text{Entropy}(\hat{\mathbf{A}}_q^{(l,h)}). \qquad (3)$$

High $H_{\text{vis}}$ signifies attentional dispersion, where the mediator fails to extract grounded evidence (further visualization in Appendix A.4).

**Joint Risk Scoring and Mediator Selection.** We define the risky mediator $H_R$ as a node that facilitates the pathological shortcut through two concurrent conditions: (1) high causal uncertainty $H_{\text{vis}}$ and (2) active prior-path transmission $m_{\text{sys}}$. The joint risk score is:

$$S^{(l,h)} = m_{\text{sys}}^{(l,h)} \cdot H_{\text{vis}}^{(l,h)}. \qquad (4)$$

This multiplicative form functions as a conjunctive filter: it selectively pinpoints mediators where the breakdown of visual grounding directly correlates with the dominance of language priors. This avoids over-penalizing heads that remain visually grounded, making the score more transferable across LVLM backbones with different attention scales. As evidenced by Fig. 3, $S$ serves as a robust diagnostic probe to localize $H_R$.

### 4.2 Reliability-Aware Causal Intervention

**RQ2.** *How can we physically block the pathological shortcut path $\mathbf{X}_{sys} \to H_R \to Y_t$ without altering fixed model parameters?*

> ***Insight II***: Effective mitigation requires a surgical intervention that severs the prior-driven shortcut while maintaining the model's structural reasoning capacity. We implement the **do**-operator via *numerical logit saturation*. By projecting the pre-softmax activations of risky mediators into a low-precision regime, we reset the pathological path to a baseline noise state without inducing the distributional shifts associated with coarse pruning.

**Causal Intervention via Numerical Saturation.** As established in § 3, the causal influence of a mediator is physically realized through its attention weights $\mathbf{A} = \text{Softmax}(\mathbf{L})$. To execute the intervention $\mathbf{do}(H_R := \text{noise})$, we modulate the logit-level parents $\mathbf{L}$ directly to achieve finer control over information flow. For any diagnosed risky mediator $(l, h) \in H_R$ at the decision-critical steps $q \in \mathcal{Q}$, we apply a substantial negative bias $\gamma$:

$$\tilde{\mathbf{L}}^{(l,h)} = \begin{cases} \Pi_{\text{dtype}}\left(\mathbf{L}^{(l,h)} - \gamma\right), & \text{if } (l,h) \in H_R, \\ \mathbf{L}^{(l,h)}, & \text{otherwise,} \end{cases} \qquad (5)$$

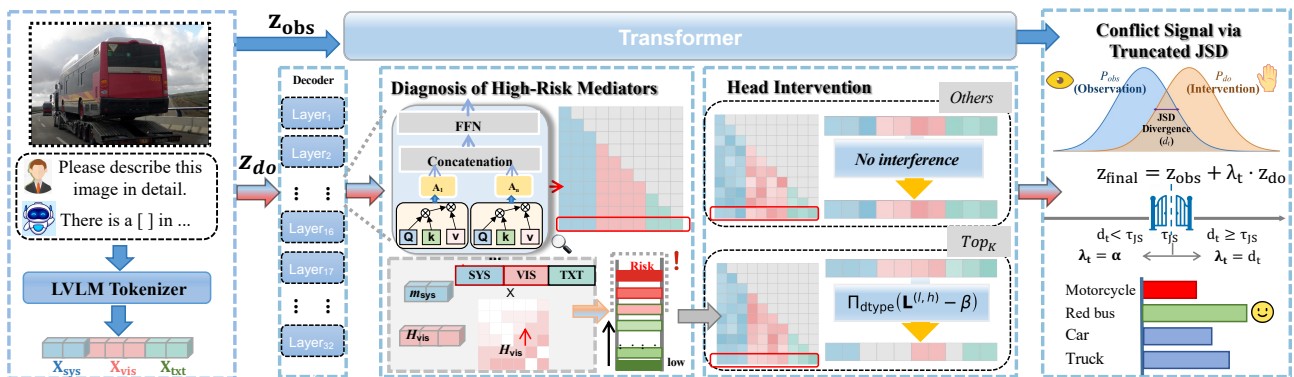

*Figure 4.* **Overview of Fox. (1) Causal Diagnosis:** Identifying risky mediators $H_R$ by intersecting decision-critical queries with the conjunctive measurement of prior-path activation $m_{\text{sys}}$ and visual uncertainty $H_{\text{vis}}$. **(2) Causal Intervention:** Executing the **do**-operator via numerical logit saturation to physically sever pathological shortcuts. **(3) Adaptive Decoding:** Reconciling observational and interventional distributions via a conflict-gated causal fusion to ensure both faithfulness and linguistic fluency.

where $\gamma$ is a large intervention constant and $\Pi_{\text{dtype}}(\cdot)$ denotes the projection onto the model's numerical precision.

**Causal Rationale of the Intervention.** While Softmax is shift-invariant in exact arithmetic, its finite-precision implementation provides a unique mechanism for path modulation. By forcing logits into a saturation regime, the intervention achieves two primary objectives:

1. *Suppression of Pathological Shortcuts*: For typical logit ranges, the exponential term $e^{x-\gamma}$ rapidly approaches the machine epsilon $\epsilon_{\text{mach}}$. This precision loss smooths out minor variances encoding spurious language priors, effectively decoupling the mediator from $\mathbf{X}_{sys}$.
2. *Preservation of Structural Anchors*: Unlike binary masking, this numerical shift allows exceptionally strong structural signals to remain distinct. If an attention peak is sufficiently robust, it can survive the precision collapse, ensuring the intervention does not destroy the essential connectivity of the model's reasoning trajectory.

The resulting distribution $\tilde{\mathbf{A}}^{(l,h)} = \text{Softmax}(\tilde{\mathbf{L}}^{(l,h)})$ represents the post-interventional state where the pathological prior-dependency is neutralized, forcing the model to rely on grounded visual trajectories.

### 4.3 Conflict-Gated Cooperative Decoding

Following the intervention in § 4.2, the model generates two concurrent distributions at each step $t$: the *observational anchor $P_{obs}$* derived from the full causal graph, and the *interventional candidate $P_{do}$* derived from $\mathbf{do}(H_R)$. While $P_{do}$ enforces visual grounding, relying on it in isolation may degrade linguistic stability. The final challenge is to adaptively reconcile these distributions to maximize faithfulness.

**RQ3.** *How can we dynamically inject interventional evidence to correct hallucinations without disrupting the model's global linguistic stability?*

> *Insight III*: Interventional signals should be calibrated by their alignment with the observational manifold. We propose *conflict-gated fusion*: when the two branches reach a consensus, we amplify the interventional signal to solidify evidence; when they diverge, we apply a conservative adjustment to preserve linguistic fluency.

**Quantifying Causal Conflict via Truncated JSD.** We measure the disagreement between the biased observation and the debiased intervention via Jensen-Shannon Divergence (JSD). To minimize tail noise, we truncate the vocabulary $\mathcal{V}$ to a candidate set $\mathcal{V}_t = \{y \in \mathcal{V} \mid P_{obs}(y) > \beta \cdot \max_w P_{obs}(w)\}$. The conflict signal $d_t$ is:

$$d_t = \text{JSD}\big(P_{obs}(\cdot|\mathcal{V}_t) \,\|\, P_{do}(\cdot|\mathcal{V}_t)\big). \quad (6)$$

Functionally, $d_t$ quantifies the sensitivity of the prediction to the pathological shortcut. Truncation reduces noise from low-probability tail tokens that fluctuate across branches without changing the selected word.

**Conflict-Gated Fusion Strategy.** Let $\mathbf{z}_{obs}$ and $\mathbf{z}_{do}$ denote the logits from the two branches. We fuse them via a dynamic weight $\lambda_t$ to obtain the final logits $\mathbf{z}_{final} = \mathbf{z}_{obs} + \lambda_t \cdot \mathbf{z}_{do}$. The weight $\lambda_t$ is governed by the conflict magnitude $d_t$ relative to a threshold $\tau_{\text{JS}}$:

$$\lambda_t = \begin{cases} \alpha, & d_t < \tau_{\text{JS}} \\ d_t, & d_t \geq \tau_{\text{JS}} \end{cases} \quad (7)$$

This strategy calibrates the interventional influence by its alignment with the observational manifold. In the consensus regime ($d_t < \tau_{\text{JS}}$), we apply a fixed gain $\alpha$ to solidify the grounded evidence. In the conflict regime ($d_t \geq \tau_{\text{JS}}$), the interventional signal acts as a *calibrated correction* where $\lambda_t = d_t$ ensures the shift toward faithfulness remains anchored to the structural stability of $P_{obs}$. The final token $y_t \sim \text{Softmax}(\mathbf{z}_{final})$ thus achieves a principled balance between interventional fidelity and linguistic fluency.

# 5 Experiments

**Models and Baselines.** We conduct experiments on three representative LVLMs: **LLaVA-1.5** (Liu et al., 2023), **Shikra** (Chen et al., 2023), and **InstructBLIP** (Dai et al., 2023). We compare Fox against five inference-time methods: **ICD** (Wang et al., 2024), **VCD** (Leng et al., 2024b), **OPERA** (Huang et al., 2024b), **SID** (Huo et al., 2025), and **CausalMM** (Zhou et al., 2025).

**Benchmarks.** We assess performance on three standard benchmarks: **POPE** (Li et al., 2023) for object existence verification; **CHAIR** (Rohrbach et al., 2018) for hallucination rates in captioning (reporting both CHAIR$_S$ and CHAIR$_I$); and **MME** (Fu et al., 2025) for comprehensive perception evaluation (reporting Accuracy and Accuracy+). Additionally, we employ **GPT-4V** as a holistic judge to assess open-ended generation quality (Huang et al., 2024b).

**Implementation Details.** We adopt a unified *Nucleus Sampling* strategy ($p = 0.9, T = 1.0$) for all models except OPERA (which requires beam search). For Fox, we fix $\alpha=2$ and $\beta=0.1$, and set $(k, \tau_{\mathrm{JS}})$ to $(0.45, 0.2)$ for LLaVA-1.5, $(0.4, 0.2)$ for InstructBLIP, and $(0.4, 0.2)$ for Shikra. All experiments are performed on NVIDIA A100 GPUs. Detailed configurations *cf.* Appendix C.2.

## 5.1 Main Results

**Results on POPE.** Table 1 reports the performance across three LVLM backbones. Fox achieves consistent improvements, notably reaching an Accuracy of 81.93% on LLaVA-1.5 in the *Adversarial* setting. This gain under high-bias conditions validates that while intensity-based methods (e.g., VCD) fail to decouple visual evidence from linguistic traps, Fox physically severs the pathological shortcut $\mathbf{X}_{sys} \to H_R \to Y_t$ via surgical intervention on risky mediators. This targeted suppression forces the model to rely on stable visual paths rather than learned co-occurrence priors. Unlike OPERA, Fox attains superior defense under standard sampling without search-based overhead, confirming that rectifying structural misalignment is more robust than heuristic-driven amplification. CausalMM achieves competitive POPE by globally adjusting attention via backdoor-based counterfactual reasoning, but its reliance on holistic causal correction without explicitly targeting a sparse set of high-risk attention heads limits its effectiveness when hallucinations are driven by localized structural shortcuts.

**Results on CHAIR.** Table 2 reports sentence-level ($C_S$) and instance-level ($C_I$) hallucination rates for long-form captioning. Fox consistently achieves the lowest scores across all backbones, notably reducing $C_I$ to 12.90 on LLaVA-1.5 and a record 11.98 on InstructBLIP. These results correspond to a relative $C_I$ reduction of 16.2% and 29.1% over SID, respectively, significantly outperforming strong baselines like OPERA and CausalMM. These results validate

| Method | LLAVA-1.5 | | | InstructBLIP | | | Shikra | | |
|---|---|---|---|---|---|---|---|---|---|
| | Ran↑ | Pop↑ | Adv↑ | Ran↑ | Pop↑ | Adv↑ | Ran↑ | Pop↑ | Adv↑ |
| Sampling | 85.13 | 82.53 | 76.77 | 86.33 | 80.53 | 77.33 | 81.63 | 80.20 | 75.57 |
| VCD | 86.84 | 80.87 | 75.33 | 85.97 | 80.63 | 78.77 | 78.13 | 80.60 | 73.80 |
| ICD | 86.27 | 83.57 | 77.90 | 87.93 | 80.47 | 77.70 | 79.07 | 79.13 | 75.87 |
| OPERA | 89.27 | **86.80** | 81.13 | 89.73 | 83.07 | **81.57** | 83.73 | **83.27** | 79.37 |
| CausalMM | 89.15 | 86.63 | 81.27 | 87.97 | 83.25 | 81.32 | 82.77 | 82.93 | 78.53 |
| SID | 88.67 | 85.37 | 81.10 | 87.43 | **83.53** | 78.60 | 82.27 | 79.20 | 79.13 |
| **Ours** | **89.33** | 86.70 | **81.93** | **89.80** | 83.40 | 80.60 | **84.97** | 82.27 | 78.77 |

*Table 1.* Results on POPE. Ran, Pop, and Adv stand for *Random*, *Popular*, and *Adversarial* settings, respectively. The higher score indicates better performance.

| Method | LLAVA-1.5 | | InstructBLIP | | Shikra | |
|---|---|---|---|---|---|---|
| | $C_S \downarrow$ | $C_I \downarrow$ | $C_S \downarrow$ | $C_I \downarrow$ | $C_S \downarrow$ | $C_I \downarrow$ |
| Sampling | 57.60 | 17.18 | 56.00 | 16.91 | 56.40 | 15.96 |
| VCD | 54.00 | 16.02 | 56.20 | 16.65 | 59.00 | 15.58 |
| ICD | 53.40 | 15.67 | 55.40 | 16.52 | 57.60 | 14.68 |
| OPERA | 49.00 | 13.52 | 45.00 | 13.15 | 53.60 | 14.20 |
| CausalMM | 47.60 | 13.49 | 43.20 | 13.05 | 53.00 | 14.27 |
| SID | 53.40 | 15.40 | 54.00 | 16.89 | 54.20 | 14.79 |
| **Ours** | **46.40** | **12.90** | **42.40** | **11.98** | **52.00** | **13.66** |

*Table 2.* Results on the CHAIR. $C_S$ and $C_I$ denote CHAIR$_S$ and CHAIR$_I$ (the smaller score indicates fewer hallucinations).

that instance-level hallucinations in descriptive tasks often stem from the pathological propagation of co-occurrence priors. By intervening on risky mediators at decision-critical steps, Fox effectively dismantles these shortcuts, forcing the model to re-verify each entity against visual evidence. The simultaneous reduction in $C_S$ and $C_I$ confirms that our conflict-gated strategy successfully rectifies structural misalignment without sacrificing descriptive richness or linguistic fluency.

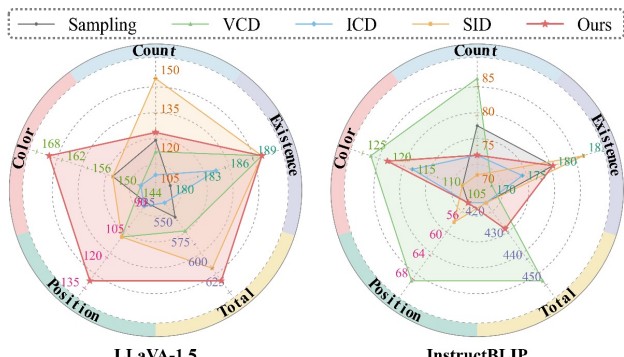

*Figure 5.* Performance on the MME benchmark. Higher scores indicate better effectiveness. Fox achieves the highest total scores across all evaluated backbones, particularly excelling in evidence-driven subsets Position and Color.

**Results on MME Benchmark.** Fig. 5 illustrates the performance across fine-grained subsets. Fox consistently improves the total score across all backbones (results for Shikra *cf.* Appendix C.6), notably increasing LLaVA-1.5's score to 613.33, surpassing the strong baseline SID (600.00). The most significant gains occur in evidence-dependent dimensions, such as *Position* ($93.33 \to 131.37$) and *Color*

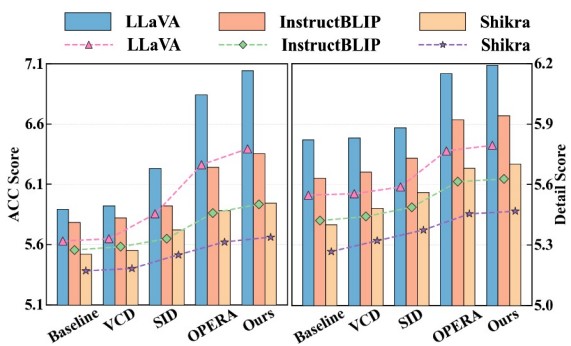

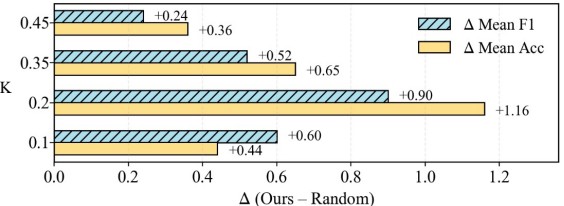

*Figure 8.* **Effectiveness of diagnosis-driven head selection.** We report the performance gains of Fox over a random-intervention baseline across different intervention ratios $K$ on the POPE benchmark. Improvements in Mean Accuracy ($\Delta$ Mean Acc) and Mean F1 ($\Delta$ Mean F1) consistently validate that targeting specific *risky mediators* is superior to stochastic head suppression.

*Figure 6.* GPT-4V-assisted hallucination evaluation. Left: Correctness (higher = less hallucination); Right: Detailedness. Lines highlight within-backbone trends.

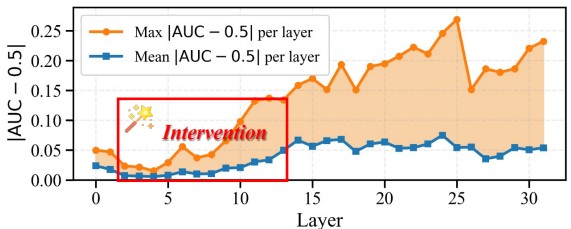

*Figure 7.* **Layer-wise diagnostic strength of visual uncertainty.** The $|\mathrm{AUC} - 0.5|$ metric (Y-axis) quantifies the separability between faithful and hallucinated samples based on entropy probes. While deeper layers exhibit higher peak discriminative strength, the red box indicates the optimal *intervention window* (early-to-mid layers) where structural rectification yields the most significant grounding improvements.

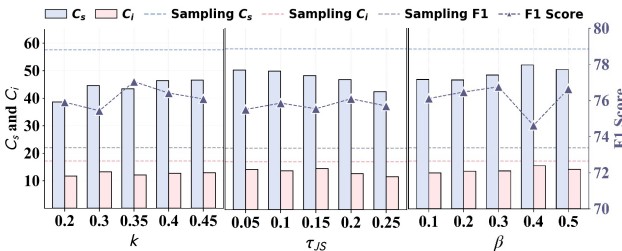

*Figure 9.* Impact of $k$, $\tau_{\mathrm{JS}}$, and $\beta$ on hallucination and informativeness in LLaVA-1.5, evaluated on 500 COCO samples.

$(150.00 \rightarrow 165.00)$. These results provide empirical proof that attribute-level assertions are highly susceptible to prior-path dominance. By disrupting the pathological shortcut at decision-critical steps, Fox forces the model to re-verify fine-grained visual evidence, explaining the marked improvements in the *Position* and *Color* subsets. The consistent enhancement across backbones confirms that rectifying structural misalignment ensures model assertions remain anchored in visual reality rather than linguistic plausibility.

**Results on GPT-4V Evaluation.** Fig. 6 summarizes the qualitative evaluation using GPT-4V as a holistic judge, following a standardized evaluation protocol in Appendix C.3. Across all backbones, Fox consistently improves *Correctness* while simultaneously enhancing *Detailedness*, indicating a superior fidelity–detail trade-off. Specifically, on LLaVA-1.5, Fox raises Correctness from 5.89 to 7.04 and Detailedness from 5.82 to 6.19. These upward trends confirm that our gains are not achieved via overly conservative or "evasive" descriptions, but via authentic visual grounding. These results further validate our causal intervention mechanism. Unlike global re-weighting strategies that often lead to distribution shifts or information loss, Fox targets risky mediators only at decision-critical steps. By surgically severing pathological shortcuts while preserving the struc-

tural integrity of the generative manifold, our method effectively suppresses prior-driven behaviors without incurring a loss in descriptive richness. Consequently, Fox rectifies the dynamic structural misalignment of baseline decoding, ensuring responses are both faithful and expressive.

### 5.2 Ablation Study

**Validation of Diagnosis and Structural Intervention.** As shown in Fig. 7, visual uncertainty signals exhibit a non-uniform distribution across layers. Although peak discriminative patterns (Max $|\mathrm{AUC} - 0.5|$) are prominent in deeper layers, we empirically find that structural intervention at these late stages yields diminishing returns for hallucination mitigation. In contrast, targeting early-to-mid layers (Layers 2–13) more effectively regulates the evidence aggregation process, achieving a superior trade-off between faithfulness and fluency. Consequently, we define this critical window as our default intervention range.

To verify the precision of our head localization, we compare diagnosis-driven selection against a *random selection* baseline under a constant budget $K$. As illustrated in Fig. 8, Fox outperforms random perturbations across $K$ values, confirming that the observed gains stem from accurately isolating a sparse set of *risky mediators* rather than stochastic noise. To further eliminate potential confounders from sampling stochasticity, we evaluate Fox in a deterministic setting (i.e., greedy decoding). In this regime, Fox maintains clear advantages over strong baselines such as VCD and

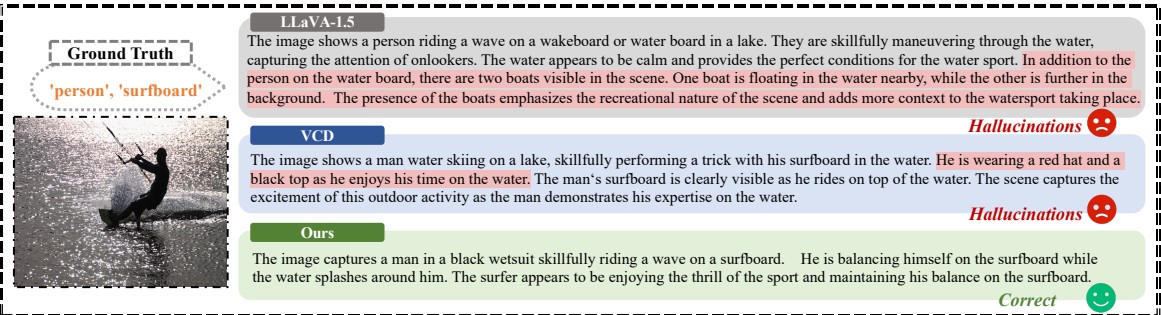

*Figure 10.* Open-ended captioning comparison. Hallucinations are marked in red. Fox effectively mitigates hallucination while preserving descriptive richness.

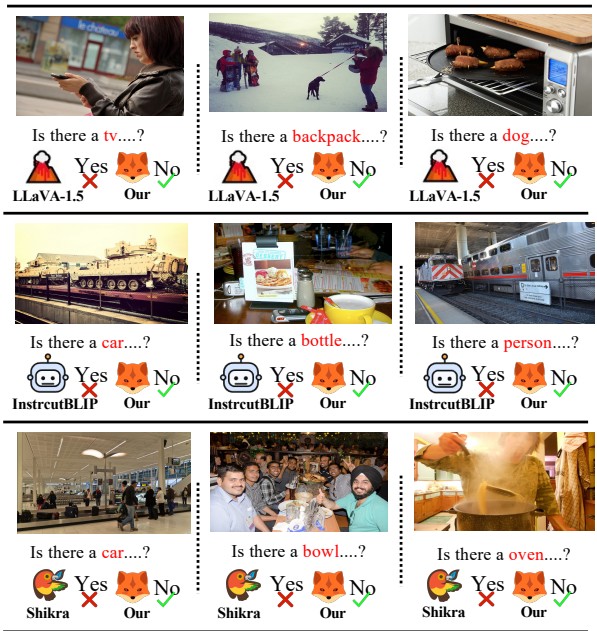

*Figure 11.* Qualitative examples of VLMs on POPE for object existence prediction. Red: Hallucination; Green: Correct predictions.

SID (details *cf.* Appendix C.7). These results reinforce that our intervention effectively addresses the dynamic structural misalignment by surgically severing pathological shortcuts.

**Hyperparameter Sensitivity.** Fig. 9 investigates the sensitivity of Fox using 500 COCO samples with LLaVA-1.5, varying the per-layer head suppression ratio $k$, the conflict threshold $\tau_{JS}$, and the truncation ratio $\beta$. Overall, Fox remains stable across a broad range of values, with $C_S$, $C_I$, and F1 changing smoothly, suggesting that the framework does not rely on fragile tuning. Among these, $\tau_{JS}$ most strongly governs the trade-off between faithfulness and informativeness. A larger $\tau_{JS}$ makes the gate more selective, suppressing prior-driven *shortcut propagation* more aggressively and typically reducing hallucination rates; however, excessive thresholds may over-constrain the generation and slightly decrease F1. In contrast, $k$ exhibits milder effects: increasing $k$ consistently lowers hallucination with minimal

F1 fluctuations. Finally, $\beta$ controls candidate truncation in conflict estimation, where moderate values yield the most robust performance. Similar trends are observed on Instruct-BLIP and Shikra (More details *cf.* Appendix C.4).

### 5.3 Case Study

Qualitative analysis confirms Fox's precision. Fig. 10 shows that in open-ended captioning, Fox restores visual grounding by suppressing prior-driven behaviors. The resulting descriptions remain strictly faithful to visual evidence without incurring a loss in descriptive richness, effectively addressing the dynamic structural misalignment inherent in standard autoregressive decoding. These examples also show that the intervention is token-selective: it does not erase the full descriptive context, but suppresses unsupported entities that emerge from prior-dominated mediators. Similarly, as illustrated in Fig. 11, while baselines succumb to existence biases by fabricating non-existent objects, Fox rectifies these errors by intervening on risky mediators to dismantle pathological shortcuts. This behavior suggests localized causal repair rather than generic conservativeness.

## 6 Conclusion & Future Work

In this paper, we challenge the prevailing attention intensity assumption and propose Fox, a training-free causal framework for mitigating hallucination by rectifying dynamic structural misalignment. By treating attention heads as dynamic mediators, Fox uses visual attention entropy to identify and suppress risky mediators that form pathological shortcuts at decision-critical steps. A conflict-gated cooperative decoding strategy further balances visual faithfulness and linguistic fluency. Extensive experiments show that Fox achieves superior hallucination reduction while preserving generation quality and maintaining the latency regime of contrastive decoding baselines. Future work will extend this causal diagnosis to compositional failures, including relation, attribute, and counting errors, and to video-language and tool-augmented LVLMs, where temporal or external evidence may introduce new shortcuts. We will also study adaptive mediator discovery under distribution shift.

## Acknowledgements

This work was supported by the Sichuan Provincial Natural Science Foundation Project (Grant No. 2026NSFSC0427), the Chengdu Technological Innovation and R&D Project (Grant No. 2026YF0800348GX), and the China Scholarship Council (CSC, Grant No. 202506070076).

## Impact Statement

This paper aims to improve the faithfulness of large vision-language models by reducing object hallucinations at inference time. More reliable multimodal generation can benefit applications such as assistive technologies, educational tools, content understanding, and information retrieval, where users may depend on model outputs to interpret visual evidence. By framing hallucination as a causal shortcut induced by dynamic mediator heads, the proposed method also provides a more interpretable perspective on when and how visual grounding fails.

At the same time, our method should not be viewed as a guarantee of factual correctness. LVLMs may still inherit dataset biases, fail under distribution shift, or produce unsupported claims in complex scenes beyond the object-centric settings studied here. Reducing hallucination may also increase users' trust in model outputs, so deployment in safety-critical or high-stakes domains should retain human oversight, uncertainty communication, and task-specific evaluation. We do not anticipate specific negative societal impacts beyond those generally associated with multimodal foundation models, and we hope this work encourages more transparent and accountable LVLM deployment.

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

# A  Additional Evidence for Motivation and Uncertain Signal

## A.1  Statistical Setup and Window Definitions

**Dataset and Sampling.** We conduct a fine-grained head-level statistical analysis on LLaVA-1.5 to provide empirical grounding for our causal assumptions. We utilize two balanced subsets: $1,000$ faithful samples and $1,000$ hallucinated samples. For each sample, we extract the raw attention probability matrices during autoregressive generation to characterize the internal information flow. We then compute metrics across the layer–head dimension to localize structural misalignment during decision-critical multimodal integration.

**Window Definitions.** To verify that our identified diagnostic signals are not transient artifacts but reflect robust structural properties of the causal graph, we define two query-position windows denoted by the length `tail`. These windows are defined on the *Text-Tail* query set $\mathcal{Q}_{\text{tail}}$, representing post-image textual positions where language priors typically begin to compete with visual evidence. The *Prefill-Last* decision point, serving as the multimodal handshake, is analyzed separately as the trajectory initialization bottleneck.

- **Instantaneous Window (tail $= 1$):** Statistics are computed using only the final query position of the current step. This setting captures the mediator behavior at the precise *instant of causal decision*, directly aligning with the "Decision-Critical Steps" identified in our SCM. It is uniquely sensitive to the transient structural "locking" where a head decouples from visual evidence to favor priors.

- **Smoothed Window (tail $= 32$):** Statistics are aggregated over the 32 most recent text query positions. This setting filters out token-level stochasticity to reveal the *persistent structural bias* of specific attention heads across a broader local context.

**Causal Complementarity.** These windows provide multi-scale validation of our framework: tail $= 1$ identifies the immediate trigger of pathological shortcuts at decision boundaries, while tail $= 32$ confirms the stability of risky mediators as unreliable information conduits. Reporting both settings demonstrates that while signal magnitude may vary, the underlying structural characteristics of *risky mediators*—specifically their tendency to mediate prior-driven influence—remain invariant to temporal smoothing.

## A.2  Global Differences in Visual Attention

We investigate whether object hallucination originates from a *quantitative* deficit in global visual grounding, as suggested by the prevailing attention intensity assumption. By analyzing the sample-level distribution of visual attention mass, we provide empirical evidence that hallucination is a structural pathology rather than a simple signal-magnitude failure.

**Head-level Visual Mass.** For each layer $l$ and head $h$, let $\mathbf{L}^{(l,h)}$ denote the pre-softmax attention activations and

$$\mathbf{A}^{(l,h)} = \text{Softmax}\Big(\mathbf{L}^{(l,h)}\Big) \tag{8}$$

denote the resulting attention weights. Given the visual token indices $\mathcal{I}_{vis} = [v_s, v_e]$ and the windowed query set $\mathcal{Q}_{\text{tail}}$ (Appendix A.1), the head-level visual attention mass is defined as:

$$m_{V,\text{tail}}^{(l,h)} = \frac{1}{|\mathcal{Q}_{\text{tail}}|} \sum_{q \in \mathcal{Q}_{\text{tail}}} \sum_{k \in \mathcal{I}_{vis}} \mathbf{A}_{q,k}^{(l,h)} \in [0,1]. \tag{9}$$

This quantity represents the total causal weight a specific mediator allocates to the visual path $X_{vis} \to H$ during the generation window.

**Sample-level Aggregation.** To evaluate the model's global observational state, we aggregate the mass across all latent mediators:

$$m_{V,\text{tail}} = \frac{1}{L \times H} \sum_{l=1}^{L} \sum_{h=1}^{H} m_{V,\text{tail}}^{(l,h)}. \tag{10}$$

**Counter-evidence for Intensity-based Explanations.** We compare the distributions of $m_{V,\text{tail}}$ for faithful and hallucinated samples in Figure 12. Critically, hallucinated outputs do not exhibit a statistically significant reduction in global visual

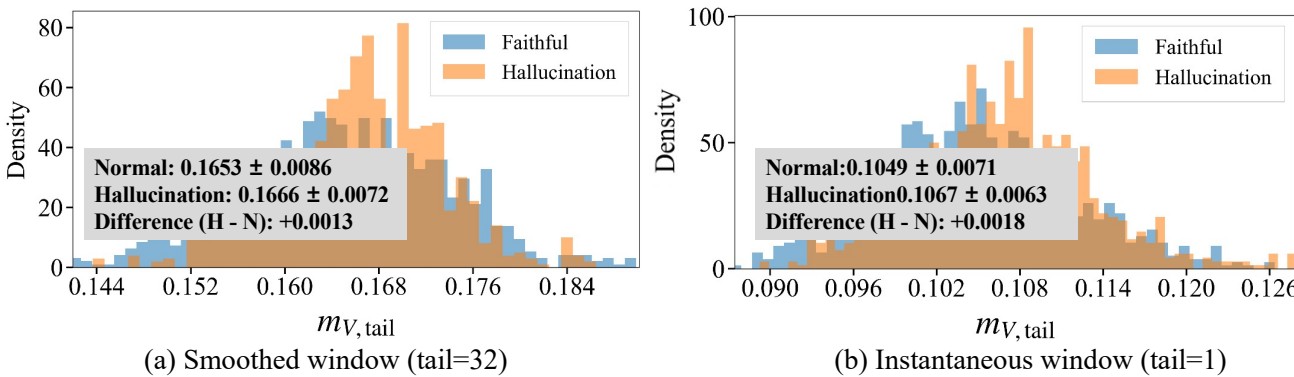

*Figure 12.* **Hallucination is not explained by a global reduction in visual attention mass.** Comparing the distribution of the sample-level global visual attention mass $m_{V,\text{tail}}$ (averaged over all layers and heads) between 1,000 Faithful (blue) and 1,000 Hallucinated (orange) samples on LLaVA-1.5-7B. **(a)** Smoothed window ($\text{tail} = 32$) aggregating statistics over recent post-image text queries. **(b)** Instantaneous window ($\text{tail} = 1$) measuring the decision-step query. The two distributions largely overlap under both windows, ruling out a coarse global-intensity account and motivating head-level structural diagnosis.

attention mass. As illustrated in Figure 12(a–b), the distributions for both groups overlap substantially across both instantaneous ($\text{tail} = 1$) and smoothed ($\text{tail} = 32$) windows.

These results provide a formal refutation of the coarse *global-intensity* hypothesis. The evidence confirms that the model maintains sufficient global visual engagement, yet fails due to *structural misalignment* at decision-critical steps. This supports our shift from global magnitude boosting toward surgical intervention on sparse, high-risk mediators that facilitate pathological shortcuts despite nominal global attention.

### A.3 Relationship between System Prompts and Hallucinations

This section provides additional evidence that reliance on system/prefix tokens is a structured signal rather than random noise, and exhibits non-trivial separability between Faithful and Hallucinated samples on our analysis set. We report results under both $\text{tail} = 1$ and $\text{tail} = 32$ windows defined in Appendix A.1, with hallucination treated as the positive class.

**System-reliance metric.** For each layer $l$ and head $h$, let $\mathbf{A}^{(l,h)} = \text{Softmax}(\mathbf{L}^{(l,h)})$ denote attention weights. Let the system/prefix indices be $\mathcal{I}_{sys} = [0, s_{img})$, where $s_{img}$ is the starting index of visual tokens. Given the windowed query set $\mathcal{Q}_{\text{tail}}$, we define the system-attention mass:

$$m_{sys,\text{tail}}^{(l,h)} = \frac{1}{|\mathcal{Q}_{\text{tail}}|} \sum_{q \in \mathcal{Q}_{\text{tail}}} \sum_{k \in \mathcal{I}_{sys}} \mathbf{A}_{q,k}^{(l,h)} \in [0, 1]. \tag{11}$$

A larger $m_{sys,\text{tail}}^{(l,h)}$ indicates stronger reliance on system/prefix priors within the observed window.

**Head-wise discriminability.** We compute the ROC-AUC for each head $(l, h)$ using $m_{sys,\text{tail}}^{(l,h)}$ as the score and report its centered value:

$$\Delta\text{AUC}^{(l,h)} = \text{AUC}^{(l,h)} - 0.5. \tag{12}$$

Here, $\Delta\text{AUC}^{(l,h)} > 0$ indicates that higher system reliance correlates with hallucination, while $\Delta\text{AUC}^{(l,h)} < 0$ indicates the opposite tendency.

**Weighted sample-level aggregation (analysis-only).** To summarize the system-reliance signal into a sample-level score for visualization, we aggregate head-wise masses with $\Delta\text{AUC}$ weighting:

$$s_n(m_{sys,\text{tail}}) = \frac{\sum_{l,h} m_{sys,\text{tail}}^{(l,h)}(n) \cdot \Delta\text{AUC}^{(l,h)}}{\sum_{l,h} \left| \Delta\text{AUC}^{(l,h)} \right|}. \tag{13}$$

This aggregation is used *only* as a diagnostic probe on the analysis set; it is not a deployed hallucination detector.

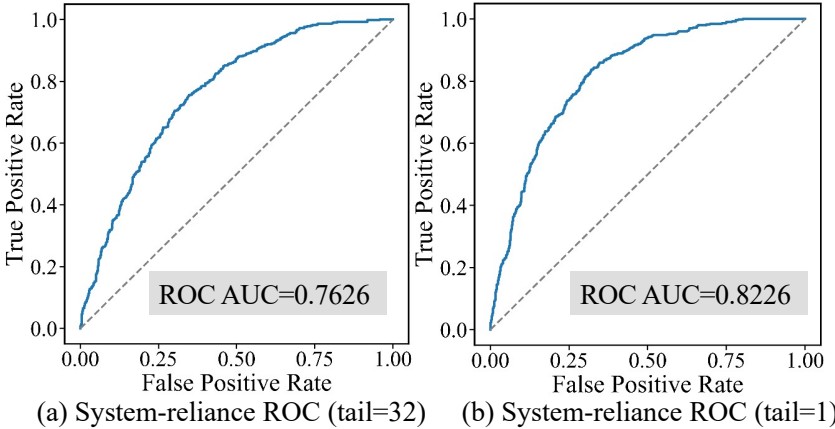

(a) System-reliance ROC (tail=32)  (b) System-reliance ROC (tail=1)

*Figure 13.* **All-head aggregated system reliance separates hallucinated from faithful samples.** We compute an analysis-only sample-level score by aggregating head-wise system attention mass $m_{sys,\text{tail}}^{(l,h)}$ over *all* heads with $\Delta\text{AUC}^{(l,h)}$ weighting (Appendix A.3); a higher score indicates stronger reliance on system/prefix tokens. **(a)** Smoothed window (tail $= 32$), AUC=0.7626. **(b)** Instantaneous window (tail $= 1$), AUC=0.8226. The stronger separability under tail $= 1$ indicates that system reliance peaks at instantaneous decision steps.

**Results and analysis.** Figure 13 plots ROC curves based on $\{s_n(m_{sys,\text{tail}})\}$. The instantaneous window (tail $= 1$) yields an AUC of 0.8226, while the smoothed window (tail $= 32$) yields an AUC of 0.7626, indicating a stable and non-trivial signal. The higher AUC under tail $= 1$ suggests that system reliance is most pronounced at the instantaneous decision step. In our method, this system-reliance cue is combined with visual uncertainty to form the joint risk score used for mediator localization in the main text.

## A.4   Visual Uncertainty and Joint Risk Scoring

This section provides complementary evidence for the joint risk formulation used in the main text (Section 4.1) to localize high-risk mediators. Rather than treating visual uncertainty as an isolated cue, we study how *visual-side instability* and *system-prior reliance* co-occur at the head level, and show that their multiplicative interaction yields a *sparse* and *stable* diagnostic signal for hallucination risk. All statistics follow the same windowed query set $\mathcal{Q}_{\text{tail}}$ and token partitions introduced earlier (Appendix A.1–A.3), with hallucination treated as the positive class.

**Visual uncertainty via modality-specific entropy.** Given attention weights $\mathbf{A}^{(l,h)}$, we quantify visual-side uncertainty by the dispersion of attention *within the visual subspace*. For each query position $q \in \mathcal{Q}_{\text{tail}}$, we re-normalize attention over visual keys $\mathcal{I}_{vis}$:

$$p_{q,k}^{(l,h)} = \frac{\mathbf{A}_{q,k}^{(l,h)}}{\sum_{j \in \mathcal{I}_{vis}} \mathbf{A}_{q,j}^{(l,h)} + \varepsilon}, \quad k \in \mathcal{I}_{vis}, \tag{14}$$

where $\varepsilon$ is a small constant for numerical stability. We then define the modality-specific visual entropy:

$$H_{vis}^{(l,h)}(q) = -\sum_{k \in \mathcal{I}_{vis}} p_{q,k}^{(l,h)} \log\left(p_{q,k}^{(l,h)} + \varepsilon\right), \tag{15}$$

and average it over the tail window:

$$H_{vis,\text{tail}}^{(l,h)} = \frac{1}{|\mathcal{Q}_{\text{tail}}|} \sum_{q \in \mathcal{Q}_{\text{tail}}} H_{vis}^{(l,h)}(q). \tag{16}$$

A larger $H_{vis,\text{tail}}^{(l,h)}$ indicates more dispersed (less certain) routing of visual evidence at decision-critical steps.

**Joint risk score for mediator localization.** Visual uncertainty alone is insufficient to characterize hallucination: a head can be visually uncertain yet benign if it does not propagate language priors into generation. Following the main text, we define a head-level joint risk score as the interaction between visual uncertainty and system reliance:

$$S_{\text{tail}}^{(l,h)} = m_{sys,\text{tail}}^{(l,h)} \cdot H_{vis,\text{tail}}^{(l,h)}. \tag{17}$$

The multiplicative form suppresses benign cases where only one factor is high and highlights heads exhibiting the co-activation pattern most consistent with prior-driven shortcut behavior.

**TopHeads-projected joint risk as a sample-level diagnostic probe.** To summarize head-wise joint risk into a sample-level score for analysis, we compute a directional weight for each head based on its head-wise ROC-AUC:

$$\Delta\text{AUC}^{(l,h)} = \text{AUC}^{(l,h)} - 0.5, \tag{18}$$

rank heads by $|\Delta\text{AUC}^{(l,h)}|$, and denote the Top-$K$ set by $\mathcal{H}_K$. For each sample $n$, we compute the weighted projection

$$s_n(S_{\text{tail}}) = \frac{\sum_{(l,h)\in\mathcal{H}_K} \widehat{S}_{\text{tail}}^{(l,h)}(n) \cdot \Delta\text{AUC}^{(l,h)}}{\sum_{(l,h)\in\mathcal{H}_K} |\Delta\text{AUC}^{(l,h)}|}, \tag{19}$$

where $\widehat{S}_{\text{tail}}^{(l,h)}(n)$ is a per-head z-score computed using dataset-level statistics (mean and standard deviation) for that head. This TopHeads projection is used *only* for diagnosing sparsity/stability on the analysis set; it does not prescribe the execution-time intervention, which follows a per-layer proportional budget in the main method.

**Sparsity and a global-aggregation reference point.** Figure 14 shows that the discriminative power of $s_n(S_{\text{tail}})$ saturates quickly as $K$ increases, indicating that the diagnostic signal is concentrated in a small subset of heads. As a reference point, if we aggregate $S_{\text{tail}}^{(l,h)}$ over *all* heads without Top-$K$ selection (i.e., replacing $\mathcal{H}_K$ by all $(l,h)$), the ROC AUC is 0.7852, which is lower than the Top-32 projection used in the main text (ROC AUC 0.8180). This suggests that global aggregation tends to dilute the localized signal rather than amplify it.

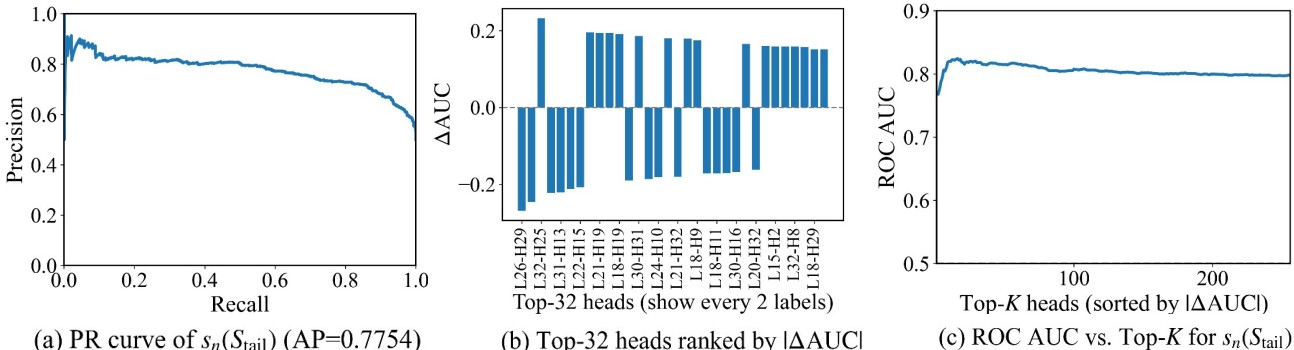

(a) PR curve of $s_n(S_{\text{tail}})$ (AP=0.7754)     (b) Top-32 heads ranked by $|\Delta\text{AUC}|$     (c) ROC AUC vs. Top-$K$ for $s_n(S_{\text{tail}})$

*Figure 14.* **Sample-level diagnosis and sparsity analysis of the joint risk score.** All statistics are computed on the *Text-Tail* query set with an instantaneous window (tail $= 1$). We define the head-level joint risk as $S_{\text{tail}}^{(l,h)} = m_{sys,\text{tail}}^{(l,h)} \cdot H_{vis,\text{tail}}^{(l,h)}$ and form a sample-level diagnostic score $s_n(S_{\text{tail}})$ by projecting standardized head-wise scores onto the Top-$K$ heads ranked by $|\Delta\text{AUC}^{(l,h)}|$. **(a)** Precision–Recall curve of the Top-32 projected joint-risk score. **(b)** The Top-32 heads ranked by $\Delta\text{AUC}^{(l,h)}$, illustrating that the signal concentrates on a sparse subset of mediators. **(c)** ROC AUC as a function of $K$, showing rapid saturation with increasing $K$.

## A.5 Visual Evidence of Uncertainty and Structural Reconfiguration Post-Intervention

In this section, we provide head-level visual evidence to support the *Risky Mediator* localization in the main text. We select representative heads ranked highly by the joint risk score $S_{\text{tail}}^{(l,h)}$ (Appendix A.4) and visualize how their attention structure changes before and after our logit-level intervention. For each head, we fix a representative sample and a decision-critical query step, and compare the Baseline (left) against the Intervention (right).

**Visualization setup.** Figure 15 provides a multi-view diagnostic snapshot for each head at the fixed query step: (i) System Reliance, the attention mass on $\mathcal{I}_{sys}$ (i.e., $m_{sys,\text{tail}}^{(l,h)}$); (ii) Visual Attention Map, the distribution over $\mathcal{I}_{vis}$ together with the corresponding visual entropy (i.e., $H_{vis,\text{tail}}^{(l,h)}$); and (iii) Text Reliance, the attention mass on the textual context. Text reliance is shown only to illustrate redistribution after intervention and is not used in risk scoring or head selection.

**Mechanistic interpretation.** Our intervention applies a large negative bias to the attention logits $\mathbf{L}^{(l,h)}$ *before* Softmax, which (under finite precision) drives unreliable connections to near-zero probability and forces the remaining mass to be re-normalized. Across the four examples, we observe a consistent reconfiguration pattern: (i) strong system/prefix lock-on is reduced; (ii) the visual map becomes less diffused and more peak-focused, reflected by a decrease in $H_{vis,\text{tail}}^{(l,h)}$; and (iii) the removed mass is redistributed to more plausible visual or textual anchors. Together, these qualitative cases provide direct evidence that heads with high $S_{\text{tail}}^{(l,h)} = m_{sys,\text{tail}}^{(l,h)} \cdot H_{vis,\text{tail}}^{(l,h)}$ indeed exhibit simultaneous system hijacking and elevated visual uncertainty, and that our intervention reconstructs their routing structure toward more grounded evidence.

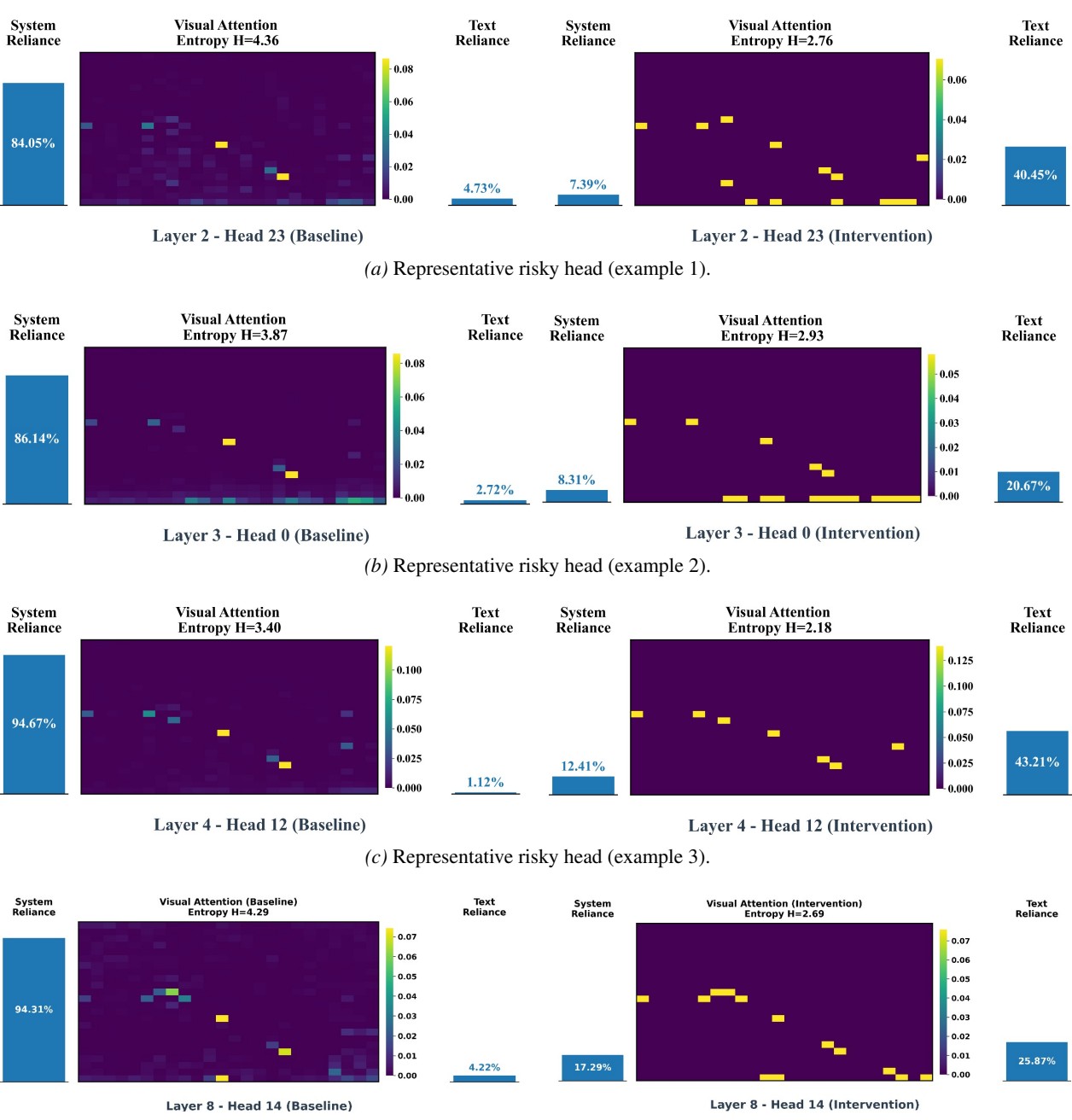

*(a)* Representative risky head (example 1).

*(b)* Representative risky head (example 2).

*(c)* Representative risky head (example 3).

*(d)* Representative risky head (example 4).

*Figure 15.* **Head-level structural transformation after logit-level intervention (four representative examples).** For each risky head (selected by high joint risk $S_{\text{tail}}^{(l,h)}$), we visualize the attention snapshot at a fixed decision-critical step: *system reliance* $m_{sys,\text{tail}}^{(l,h)}$ (left bar), *visual attention map* over $\mathcal{I}_{vis}$ with entropy $H_{vis,\text{tail}}^{(l,h)}$ (center heatmap), and *text reliance* (right bar). Across examples, the intervention consistently suppresses pathological system/prefix lock-on and sharpens visual evidence routing, leading to a lower visual entropy and a more concentrated set of visual peaks. These four cases illustrate that the structural reconfiguration induced by our intervention is not confined to a single head or layer, but recurs across multiple risky mediators.

# B  Method

This subsection specifies how Fox *executes* the causal intervention at inference time. After identifying the risky mediators $H_R$ in our diagnosis stage, we instantiate two coupled next-token distributions at each decoding step $t$: (i) an *observational* branch $P_{obs}(\cdot)$ that follows the original computation graph, and (ii) an *interventional* branch $P_{do}(\cdot)$ that enforces $\mathbf{do}(H_R)$

by editing attention logits before Softmax on the selected heads. Causally, the intervention aims to attenuate the shortcut pathway $\mathbf{X}_{sys} \to H_R \to Y_t$ so that the next-token decision relies less on system/prefix priors and becomes more grounded in the multimodal evidence, while the observational branch preserves the model's native linguistic manifold. Our objective is to leverage the improved faithfulness of $P_{do}$ without over-committing to it when the intervention becomes overly restrictive.

**Step-wise procedure.** Concretely, at each generation step, we perform the following operations:

- **(1) Dual forward passes (observational vs. interventional).** We compute the next-token logits from the original run to obtain $P_{obs}$. In parallel, we run the model again with $\mathbf{do}(H_R)$ applied on the diagnosed heads to obtain $P_{do}$. In practice, $\mathbf{do}(H_R)$ is implemented exactly as in Algorithm 1: on the designated intervention layers (early-to-mid range), we select Top-$\lceil k \cdot H \rceil$ heads per layer using the joint risk score and apply a negative bias to their *pre-Softmax* attention logits on decision-critical queries, driving unreliable links to near-zero probability after re-normalization. Concretely, for $(l, h) \in H_R$ and $q \in \mathcal{Q}$, we use numerical logit saturation: $\tilde{\mathbf{L}}^{(l,h)} = \Pi_{\text{dtype}}(\mathbf{L}^{(l,h)} - \gamma)$, and then $\tilde{\mathbf{A}}^{(l,h)} = \text{Softmax}(\tilde{\mathbf{L}}^{(l,h)})$.

- **(2) Candidate truncation for stable conflict measurement.** Measuring divergence over the full vocabulary is dominated by the long tail of near-zero probabilities. We therefore construct a decision-relevant candidate set $\mathcal{V}_t$ induced by $P_{obs}$, retaining only tokens within a fixed ratio of the top-1 probability. This truncation makes the conflict estimate focus on the local decision boundary rather than numerical tail noise.

- **(3) Conflict estimation.** We quantify the disagreement between the observational and interventional branches using Jensen–Shannon divergence on the truncated candidates:

$$d_t = \text{JSD}(P_{obs}(\cdot \mid \mathcal{V}_t) \,\|\, P_{do}(\cdot \mid \mathcal{V}_t)). \tag{20}$$

  A small $d_t$ indicates that the intervention stays close to the observational manifold at step $t$, whereas a large $d_t$ signals a strong causal perturbation that substantially reshapes the next-token preference.

- **(4) Conflict-gated injection.** We convert $d_t$ into a step-wise injection weight $\lambda_t$. When the two branches are consistent (low conflict), we apply a fixed gain $\alpha$ to strengthen the interventional correction. When they diverge (high conflict), we fall back to a softer, conflict-proportional injection, preventing the interventional branch from overwhelming the observational manifold.

**Logit-level combination.** Finally, we couple the two branches at the logit level and select the next token:

$$\mathbf{z}_{final,t} = \mathbf{z}_{obs,t} + \lambda_t \cdot \mathbf{z}_{do,t}, \qquad y_t \sim \text{Softmax}(\mathbf{z}_{final,t}), \tag{21}$$

where $\lambda_t$ is determined by the conflict-gating rule described above.

---

**Algorithm 1: Fox framework**

**Input:** $\mathbf{X}$, history $y_{<t}$, model $\mathcal{F}_\theta$; intervention layers $\mathcal{L}_{int}$; decision queries $\mathcal{Q}$; per-layer ratio $k$; intervention bias $\gamma$; conflict threshold $\tau_{\text{JS}}$; gain $\alpha$; truncation ratio $\beta$.

**Output:** next token $y_t$ (repeat for $t = 1, \ldots, T$).

**(1) Pass-1.** Compute logits $\mathbf{z}_t^{obs} = \mathcal{F}_\theta(\mathbf{X}, y_{<t})$ and $P_{obs} = \text{Softmax}(\mathbf{z}_t^{obs})$.

**(2) Pass-2 (with intervention on $\mathcal{L}_{int}$).** Run $\mathcal{F}_\theta(\mathbf{X}, y_{<t})$ again; for each layer $l \in \mathcal{L}_{int}$:

    (a) On decision queries $\mathcal{Q}$, compute per-head metrics $m_{sys}^{(l,h)}$ and $H_{vis}^{(l,h)}$, and score $S^{(l,h)} = m_{sys}^{(l,h)} \cdot H_{vis}^{(l,h)}$.

    (b) Select risky heads $\mathcal{H}_l = \text{Top-}\lceil k \cdot N_h \rceil$ by $S^{(l,h)}$.

    (c) **Pre-Softmax logit intervention:** for $h \in \mathcal{H}_l$ and $q \in \mathcal{Q}$, apply a negative bias $-\gamma$ to the corresponding attention logits.

Obtain interventional logits $\mathbf{z}_t^{do}$ and $P_{do} = \text{Softmax}(\mathbf{z}_t^{do})$.

**(3) Conflict gating.** Construct $\mathcal{V}_t = \{y \in \mathcal{V} \mid P_{obs}(y) \geq \beta \cdot \max_w P_{obs}(w)\}$ and compute

$$d_t = \text{JSD}(P_{obs}(\cdot \mid \mathcal{V}_t) \,\|\, P_{do}(\cdot \mid \mathcal{V}_t)).$$

Set $\lambda_t = \alpha$ if $d_t < \tau_{\text{JS}}$, otherwise $\lambda_t = d_t$.

**(4) Logit fusion and selection.** Fuse $\mathbf{z}_{final,t} = \mathbf{z}_{obs,t} + \lambda_t \cdot \mathbf{z}_{do,t}$ and select $y_t \sim \text{Softmax}(\mathbf{z}_{final,t})$.

---

Algorithm 1 summarizes the step-wise inference procedure.

# C  Detailed Configurations and Experimental Results

## C.1  Models and Baselines

We select LVLMs that represent diverse architectural paradigms to ensure the generalizability of our method:

- **LLaVA-1.5** (Liu et al., 2023): A widely-used general-purpose baseline that connects a CLIP-ViT-L/14 encoder with the Vicuna LLM via a two-layer MLP projection.

- **Shikra** (Chen et al., 2023): A structured LVLM specialized for referential dialogue and fine-grained object grounding, handling bounding box inputs/outputs.

- **InstructBLIP** (Dai et al., 2023): An instruction-tuned model utilizing a Q-Former to compress visual features into soft queries for the LLM.

For baselines, we compare against the following inference-time interventions:

- **ICD** (Wang et al., 2024): Constructs contrastive instruction branches to estimate and suppress language priors.

- **VCD** (Leng et al., 2024b): Introduces visual noise to amplify hallucination-prone logits via contrastive decoding.

- **OPERA** (Huang et al., 2024b): A beam-search-based method that detects "over-trust" attention patterns and applies a rollback penalty.

- **SID** (Huo et al., 2025): Reweights candidate tokens dynamically using self-contrastive signals to prevent error amplification.

- **CausalMM** (Zhou et al., 2025): Applies counterfactual reasoning on both encoder and decoder sides to disentangle spurious correlations.

## C.2  Hyperparameters and Hardware

**Sampling Strategy.** To simulate realistic generation scenarios, we use Nucleus Sampling with top-$p = 0.9$, temperature $T = 1.0$, and a maximum length of 512 tokens. No repetition or length penalties are applied. Note that OPERA is evaluated using its official beam-search configuration (num_beams=5) as it is incompatible with standard sampling.

**Method-Specific Parameters.** Our method involves four key hyperparameters: the per-layer head suppression ratio $k$, the conflict threshold $\tau_{\mathrm{JS}}$, the consensus amplification factor $\alpha$, and the truncation ratio $\beta$ used in conflict estimation.

We fix $\alpha = 2$ across all backbones. The remaining hyperparameters $(k, \tau_{\mathrm{JS}}, \beta)$ are selected in a model-specific manner via grid search on a held-out validation set. Notably, the search consistently selects the same $\beta$ across all evaluated backbones, and we therefore use $\beta = 0.1$ for LLaVA-1.5, InstructBLIP, and Shikra.

The optimal configurations are:

- **LLaVA-1.5:** $k = 0.45$, $\tau_{\mathrm{JS}} = 0.2$, $\beta = 0.1$.

- **InstructBLIP:** $k = 0.4$, $\tau_{\mathrm{JS}} = 0.2$, $\beta = 0.1$.

- **Shikra:** $k = 0.4$, $\tau_{\mathrm{JS}} = 0.2$, $\beta = 0.1$.

We report the best results averaged over 10 independent runs. Statistical significance is determined by a two-sided $t$-test ($p < 0.05$). All experiments are conducted on $8\times$ NVIDIA A100 (40GB) GPUs using PyTorch and HuggingFace Transformers.

## C.3  Detailed Metrics and Protocols

**(1) POPE.** This metric evaluates object existence through a series of binary (Yes/No) questions (e.g., "*Is there a [object] in the image?*"), thereby measuring the model's propensity to fabricate non-existent visual evidence. POPE consists of three distinct sampling configurations to assess different facets of model reliability:

- **Random:** Targets are sampled randomly with broad category coverage to examine the model's general recognition and grounding capabilities across diverse objects.

- **Popular:** Categories with higher frequencies in the training distribution are selected to observe if the model is more stable and less prone to hallucination when dealing with "common and familiar" objects.

- **Adversarial:** Categories that are frequently misreported or confused by LVLMs are selected, increasing the evaluation difficulty and revealing the model's robustness against hallucination under ambiguous or interfering inputs.

By focusing on binary probing, this benchmark bypasses the complexities associated with parsing open-ended generated captions, ensuring a stable, fair, and adaptable evaluation process. We report both Accuracy and the F1-score to quantify performance. The F1-score serves as a balanced harmonic mean between Precision and Recall, defined as:

$$\text{Recall} = \frac{\text{Correctly identified objects}}{\text{Ground-truth objects}}, \tag{22}$$

$$\text{Precision} = \frac{\text{Correctly identified objects}}{\text{Total generated objects}}, \tag{23}$$

$$\text{F1} = 2 \times \frac{\text{Precision} \times \text{Recall}}{\text{Precision} + \text{Recall}}. \tag{24}$$

In our experimental framework, Recall characterizes the proportion of ground-truth objects successfully retrieved by the model from the visual evidence $\mathbf{X}_{vis}$, while Precision measures the ratio of generated objects that actually exist in the image rather than being hallucinations. As the harmonic mean of both, the F1-score provides a robust holistic measure of generation quality. Consequently, Accuracy and F1-score constitute our standard baseline framework for assessing the model's overall efficacy in multimodal grounding.

**(2) CHAIR.** The **CHAIR** (Rohrbach et al., 2018) benchmark consists of two primary metrics, $\text{CHAIR}_I$ and $\text{CHAIR}_S$, which measure object hallucinations in image captioning at the instance and sentence levels, respectively. Specifically, the instance-level metric $\text{CHAIR}_I$ calculates the proportion of hallucinated objects relative to all mentioned objects in the generated captions. The sentence-level metric $\text{CHAIR}_S$ reflects the proportion of generated sentences that contain at least one hallucination.

To ensure that our intervention $\mathbf{do}(H_R)$ does not merely reduce hallucinations by excessively suppressing the generation of fine-grained details, we further incorporate Recall and F1-score as indicators of semantic completeness. These metrics verify that the performance gains are not achieved through an "evasion strategy" but through reliable visual grounding. The metrics follow the formulations below:

$$\text{CHAIR}_I = \frac{|\{\text{hallucinated objects}\}|}{|\{\text{all mentioned objects}\}|}, \tag{25}$$

$$\text{CHAIR}_S = \frac{|\{\text{sentences with hallucinations}\}|}{|\{\text{total sentences}\}|}. \tag{26}$$

The robustness of the captions is measured via:

$$\text{Recall} = \frac{|\{\text{accurately mentioned objects}\}|}{|\{\text{ground-truth objects}\}|}, \tag{27}$$

$$\text{Precision} = \frac{|\{\text{all mentioned objects}\} \cap \{\text{ground-truth objects}\}|}{|\{\text{all mentioned objects}\}|}, \tag{28}$$

$$\text{F1} = 2 \times \frac{\text{Precision} \times \text{Recall}}{\text{Precision} + \text{Recall}}. \tag{29}$$

Collectively, $\text{CHAIR}_I$, $\text{CHAIR}_S$, and the F1-score constitute our comprehensive evaluation framework for the captioning task.

**(3) MME.**The **MME** (Fu et al., 2025) benchmark pairs each image with two semantically similar questions whose ground-truth answers are "Yes" and "No," respectively. Evaluation is conducted using two metrics: Accuracy and Accuracy+.

- **Accuracy** is calculated at the question granularity: a correct response to any single question contributes to the score.

- **Accuracy+** is calculated at the image granularity: a sample is counted as correct only if the model correctly answers *both* the "Yes" and "No" questions associated with the same image.

This stricter requirement ensures that the model truly perceives the visual evidence $\mathbf{X}_{vis}$ rather than relying on language priors. The final MME Score is defined as the sum of these metrics:

$$\text{Accuracy} = \frac{\sum_{i \in \mathcal{I}} \mathbf{1}[f(i, q_{yes}) = \text{"Yes"}] + \sum_{i \in \mathcal{I}} \mathbf{1}[f(i, q_{no}) = \text{"No"}]}{2|\mathcal{I}|}, \tag{30}$$

$$\text{Accuracy}^+ = \frac{\sum_{i \in \mathcal{I}} \mathbf{1}[f(i, q_{yes}) = \text{"Yes"} \wedge f(i, q_{no}) = \text{"No"}]}{|\mathcal{I}|}, \tag{31}$$

$$\text{MME Score} = \text{Accuracy} + \text{Accuracy}^+. \tag{32}$$

**(4) GPT-4V Assisted Evaluation.**While CHAIR and POPE effectively identify object hallucinations, they provide limited insight into the overall linguistic quality and descriptive richness of open-ended outputs. To complement these automatic metrics, we conduct a GPT-4V–assisted evaluation by following an established protocol from prior work (Huang et al., 2024b; Yang et al., 2023). Specifically, GPT-4V acts as a multimodal judge and assigns scores under two criteria: Accuracy (factual consistency with respect to objects, attributes, and spatial/relational correctness) and Detailedness (the richness and precision of *correctly grounded* visual details). Leveraging its advanced perception, GPT-4V can capture subtle errors in color, spatial positioning, and logical relationships between objects. We conduct this evaluation on a curated subset of 50 images from the MS-COCO 2014 validation set, used solely as complementary evidence rather than a primary benchmark. For each image–model pair, we generate descriptions with a standardized prompt and evaluate both the original backbone output and the output produced with our method. GPT-4V inference is performed with max_tokens=512 and temperature=0.2. The full prompt and required output format are provided in Table 3.

### C.4    Additional Ablation Studies

**Efficiency and Latency Analysis.** Table 4 compares the inference cost on LLaVA-1.5 alongside POPE Adversarial performance. Fox achieves a superior Pareto trade-off, maintaining the same latency regime as VCD/SID ($\approx$200 ms/token) while reaching the highest accuracy (81.93%). For 10-token generation, Fox incurs only modest overhead ($1,040$ ms), whereas search-based methods like OPERA exhibit significantly higher latency ($2,560$ ms) due to beam search and iterative rollback. These results confirm that our gains stem from a logit-level intervention within the same dual-pass contrastive decoding regime as VCD/SID, rather than increased decoding depth or search budgets. In practice, the system-prompt and visual-token KV cache can be shared between branches, keeping the overhead in the same latency class while providing a stronger balance between faithfulness and efficiency.

### C.4.1    ABLATION STUDY ON $\alpha$

As shown in Table 5, we conduct a sensitivity analysis of the enhancement factor $\alpha$ on POPE to determine a default setting that can be reused across backbones. Concretely, we vary $\alpha$ on LLaVA-1.5 while keeping all other hyperparameters fixed, and report the averaged *Accuracy* and *F1* over the Random/Popular/Adversarial splits. We observe that performance changes smoothly within a reasonably wide range of $\alpha$, with $\alpha = 2$ achieving the best or near-best averaged performance. When further increasing $\alpha$, the gains exhibit diminishing returns and slightly regress on some splits, suggesting that overly strong consensus enhancement may bias the binary verification toward a more *robust but conservative* decision behavior, which is detrimental to overall F1. Based on this trend, we set $\alpha = 2$ as the default and keep it fixed for all subsequent backbones (including InstructBLIP and Shikra), reducing model-specific tuning freedom and verifying the transferability and robustness of this choice.

| GPT-4V Prompt |
|---|

**GPT-4V Prompt**

You are required to score the performance of two AI assistants in describing a given image. You should pay extra attention to the hallucination, which refers to the part of descriptions that are inconsistent with the image content, such as claiming the existence of something not present in the image or describing incorrectly in terms of the counts, positions, or colors of objects in the image. Please rate the responses of the assistants on a scale of 1 to 10, where a higher score indicates better performance, according to the following criteria:

**1. Accuracy:** Evaluate whether the response is accurate and faithful to the actual image content. Focus on identifying any hallucinations including non-existent objects, incorrect attributes (colors, sizes, materials), wrong quantities, false spatial relationships, or activities that are not happening. Responses with fewer hallucinations and higher fidelity to the image should receive higher scores.

**2. Detailedness:** Assess whether the response provides rich and informative details about the image. Consider the completeness of the description, coverage of important visual elements, and the depth of observations. Note that hallucinated content does NOT count as valid details – only accurate information contributes to this score.

Please output the scores for each criterion, containing only two values indicating the scores for Assistant 1 and 2, respectively. The two scores are separated by a space. Following the scores, please provide an explanation of your evaluation, avoiding any potential bias and ensuring that the order in which the responses were presented does not affect your judgment.

[Assistant 1]
{Response of Assistant 1}
[End of Assistant 1]
[Assistant 2]
{Response of Assistant 2}
[End of Assistant 2]

**Output format:**
Accuracy: <score_1> <score_2>
Reason: <your explanation>
Detailedness: <score_1> <score_2>
Reason: <your explanation>

*Table 3.* The prompt used for GPT-4V evaluation.

### C.4.2    ABLATION STUDY ON INSTRUCTBLIP

We analyze the sensitivity of key hyperparameters on InstructBLIP. Since $\alpha$ is fixed globally to $\alpha = 2$ across all backbones (see Appendix C.4.1), we focus on the three parameters that govern intervention strength and conflict gating: the per-layer head suppression ratio $k$, the JSD conflict threshold $\tau_{\text{JS}}$, and the truncation ratio $\beta$. Overall, these parameters mainly control the trade-off between *reliability* (lower hallucination, reflected by CHAIR$_S$/CHAIR$_I$) and *informativeness* (semantic coverage, reflected by F1), with $\tau_{\text{JS}}$ serving as the primary knob.

**Impact of $\tau_{\text{JS}}$.** As $\tau_{\text{JS}}$ increases, the decoding procedure more frequently follows the intervention-dominant path at uncertain steps, which consistently reduces hallucination errors (lower CHAIR$_S$ and CHAIR$_I$). However, overly large $\tau_{\text{JS}}$ yields diminishing returns and may slightly regress F1, indicating a conservative bias that prioritizes safety over semantic coverage. This suggests that $\tau_{\text{JS}}$ should be set within a moderate range to suppress uncertainty-driven deviations without over-stabilizing the output.

**Impact of $k$.** The ratio $k$ controls the sparsity budget of head-level intervention per layer. Varying $k$ results in relatively smooth changes in both CHAIR and F1: moderate $k$ values typically improve CHAIR$_S$/CHAIR$_I$ without harming F1, while overly aggressive intervention (large $k$) brings limited additional gains and can slightly reduce F1, suggesting over-contraction of the effective generation space.

**Impact of $\beta$.** The parameter $\beta$ controls the truncation strength in conflict estimation. InstructBLIP is relatively robust to $\beta$ within a broad range, where performance varies smoothly and improvements on CHAIR metrics remain stable. Extreme truncation may narrow the effective candidate space and lead to diminishing returns, occasionally accompanied by a slight F1 decrease.

| Method | 1/token (ms) | 10/token (ms) | ACC (%) |
|--------|--------------|---------------|---------|
| Sampling | 110 | 437 | 76.77 |
| VCD | 204 | 893 | 75.33 |
| OPERA | - | 2560 | 81.13 |
| SID | 202 | 890 | 81.10 |
| **Ours** | 206 | 1040 | **81.93** |

*Table 4.* Efficiency comparison. Inference latency (ms) for 1-token and 10-token generation. OPERA is not evaluated in the 1-token setting (marked as -).

| $\alpha$ | Mean Acc↑ | Mean F1↑ |
|----------|-----------|----------|
| 0 | 81.75 | 82.90 |
| 1 | 85.33 | 85.35 |
| 2 | 85.97 | **85.77** |
| 3 | **85.98** | 85.53 |

*Table 5.* Sensitivity of the enhancement factor $\alpha$ on POPE.

**Selection of Intervention Layers in InstructBLIP.** For InstructBLIP, we set the intervention range to layers 4–10, adhering to our core finding that intervention must occur in the early-to-mid stages to suppress *risky mediators* before erroneous evidence chains become consolidated in subsequent generation.

The distinction lies in InstructBLIP's architecture: cross-modal information is first compressed into a compact set of visual representations via the Q-Former's learnable queries before fusion with the LLM. Consequently, the "effective visual evidence" enters the language decoder in a manner that favors early information aggregation followed by progressive verbalization.

- **Early Layers** ($< 4$)**:** Intervening too early often acts on the stage before multimodal fusion is fully realized, resulting in unstable gains.

- **Latter Layers** ($> 10$)**:** Intervening in later stages primarily affects linguistic expression and decoding convergence, offering limited help in correcting biases formed during the initial fusion stage, and potentially leading to overly conservative outputs.

By targeting the 4–10 layer range, we effectively balance the suppression of uncertainty-driven shortcuts with the preservation of the model's inherent linguistic fluency.

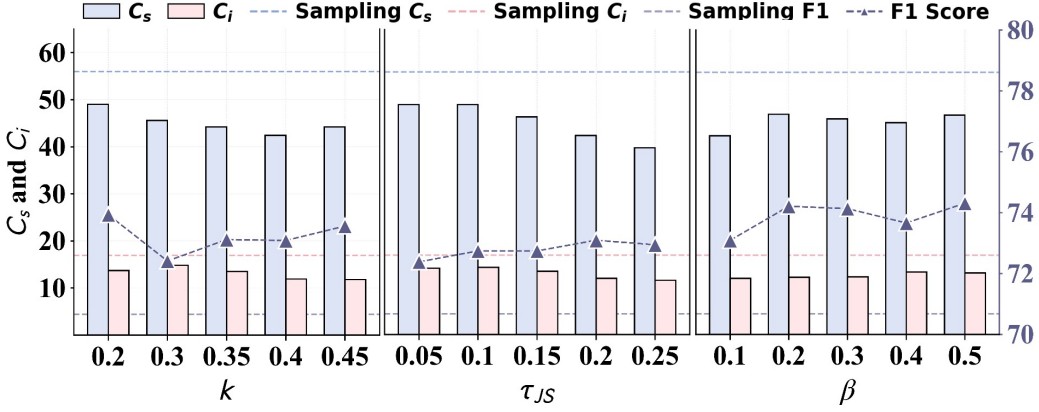

*Figure 16.* Parameter sensitivity analysis on InstructBLIP. Impact of $k$, $\tau_{\text{JS}}$, and $\beta$ on captioning performance (CHAIR$_S$, CHAIR$_I$, and F1), evaluated on 500 COCO samples.

### C.4.3 Ablation Studies on Shikra

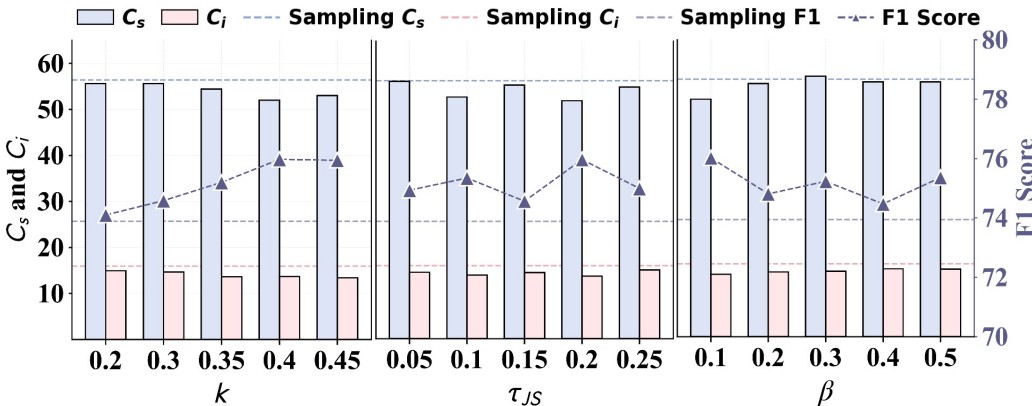

*Figure 17.* Parameter sensitivity analysis on Shikra. Impact of $k$, $\tau_{\mathrm{JS}}$, and $\beta$ on captioning performance (CHAIR$_S$, CHAIR$_I$, and F1), evaluated on 500 COCO samples.

As shown in Figure 17, we examine the sensitivity of key hyperparameters on Shikra. Since $\alpha$ is fixed globally to $\alpha = 2$ across all backbones (see Appendix C.4.1), we focus on the three parameters that directly govern intervention strength and conflict gating: the per-layer head suppression ratio $k$, the JSD conflict threshold $\tau_{\mathrm{JS}}$, and the truncation ratio $\beta$. Overall, these parameters mainly control the trade-off between *reliability* (lower hallucination, reflected by CHAIR$_S$/CHAIR$_I$) and *informativeness* (semantic coverage, reflected by F1), with $\tau_{\mathrm{JS}}$ remaining the most influential knob.

**Impact of $\tau_{\mathrm{JS}}$.** As $\tau_{\mathrm{JS}}$ increases, the decoding procedure more often follows the intervention-dominant path at uncertain steps, yielding consistent reductions in hallucination metrics (lower CHAIR$_S$ and CHAIR$_I$). However, overly large $\tau_{\mathrm{JS}}$ introduces diminishing returns and may slightly reduce F1, indicating a conservative bias that favors safer generation over semantic coverage. This suggests that $\tau_{\mathrm{JS}}$ should be set in a moderate range to suppress uncertainty-driven deviations without over-stabilizing the output.

**Impact of $k$.** The ratio $k$ controls the sparsity budget of head-level intervention per layer. Figure 17 shows that varying $k$ leads to relatively smooth changes in both CHAIR and F1. Moderate $k$ values typically provide a favorable balance, improving CHAIR$_S$/CHAIR$_I$ without harming F1, while overly aggressive intervention (large $k$) offers limited additional gains and can slightly regress F1.

**Impact of $\beta$.** The parameter $\beta$ controls the truncation strength in conflict estimation. Performance varies smoothly across a broad range of $\beta$, suggesting that Shikra is relatively robust to this parameter. Moderate $\beta$ values achieve stable improvements on CHAIR metrics while maintaining F1, whereas extreme truncation may narrow the effective candidate space and yield diminishing returns.

**Selection of Intervention Layers in Shikra.** For Shikra, we set the intervention range to layers 3–10. The core rationale remains consistent with our previous findings: the erroneous evidence chains of hallucinations are typically established and progressively amplified during the early-to-mid stages of decoding. Intervening only in the later stages is usually insufficient for timely correction. Therefore, we place the intervention within the "early-to-mid" window to suppress the influence of risky mediators on subsequent generation as early as possible.

Compared to LLaVA-1.5 (e.g., layers 2–15), Shikra's optimal window is slightly shifted forward and is narrower, primarily due to differences in model architecture and task format.

- **Early Structural Routing:** As a model designed for referential grounding, Shikra's input sequences contain more explicit region/object-related tokens. Cross-modal alignment forms strong structural routing in earlier layers.

- **Generative Convergence:** Subsequent layers tend toward the "convergence" of linguistic generation and instruction execution based on the established alignment. Intervening at this stage is more likely to disrupt stable generation while providing limited help in correcting errors formed during early alignment.

Based on this divergence, while maintaining the principle of "early-to-mid stage intervention," we set Shikra's intervention

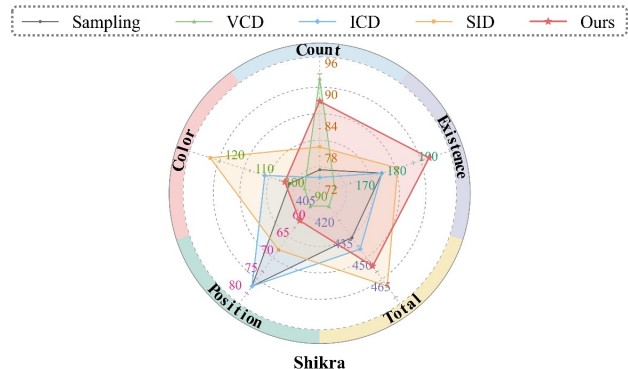

*Figure 18.* Performance on the MME benchmark. Higher scores indicate better effectiveness.

range to layers 3–10 to better fit its internal dynamics where alignment is established early and generation converges later.

| Method | LLAVA-1.5 | | | InstructBLIP | | | Shikra | | |
|---|---|---|---|---|---|---|---|---|---|
| | Ran↑ | Pop↑ | Adv↑ | Ran↑ | Pop↑ | Adv↑ | Ran↑ | Pop↑ | Adv↑ |
| Sampling | 85.59 | 83.40 | 79.06 | 86.14 | 81.55 | 78.80 | 82.17 | 81.06 | 77.44 |
| VCD | 86.83 | 82.05 | 78.00 | 85.70 | 81.12 | 79.87 | 79.13 | 81.12 | 75.90 |
| ICD | 86.46 | 84.18 | 79.83 | 87.90 | 81.74 | 79.82 | 80.14 | 80.20 | 77.89 |
| OPERA | 88.72 | **86.59** | 81.87 | 89.43 | 83.19 | **82.34** | 83.66 | **83.21** | **80.01** |
| CausalMM | 88.63 | 86.17 | 81.93 | 87.83 | 83.27 | 82.23 | 82.53 | 82.89 | 79.47 |
| SID | 88.18 | 85.21 | 81.69 | 86.10 | 83.51 | 80.34 | 82.28 | 79.97 | 80.20 |
| **Fox (Ours)** | **88.74** | 86.30 | **82.26** | **89.44** | **83.77** | 81.56 | **84.68** | 82.27 | 79.55 |

*Table 6.* POPE F1 Score on the Random/Popular/Adversarial splits for three LVLM backbones (LLaVA-1.5, InstructBLIP, and Shikra). Higher is better.

## C.5 POPE F1 Results

In the main paper, we report Accuracy on POPE as the primary metric to emphasize hallucination suppression. To provide a complementary view of the precision–recall trade-off, we additionally report F1 scores on all POPE splits in Table 6. Overall, our method remains *competitive* in F1 across backbones and splits: it achieves the best F1 on LLaVA-1.5 (Random/Adversarial) and on InstructBLIP (Random/Popular), while maintaining comparable performance on the remaining settings. These results suggest that the accuracy gains reported in the main paper are not obtained by a degenerate conservative strategy that trivially avoids positive answers, but are accompanied by a balanced precision–recall behavior.

## C.6 Additional MME Results

As shown in Figure 18, in SHIKRA, Fox still achieves an overall improvement on the MME benchmark, increasing the total score from 430.00 to 446.63, which validates the effectiveness of our structural intervention under this backbone. Specifically, Fox yields stable gains on object-level dimensions, including *Existence* (180 → 195) and *Count* (75 → 90), and also brings a slight improvement on *Color* (96.67 → 98.33). In contrast, *Position* exhibits a noticeable drop (78.33 → 63.33), suggesting that Shikra is more sensitive to attention-structure changes for spatial attribute verification, and the benefits of structural intervention can vary across fine-grained dimensions.

## C.7 Results under Greedy Decoding

To rule out potential confounding effects introduced by stochastic decoding (e.g., sampling temperature or nucleus sampling), we further evaluate all methods under greedy decoding on LLaVA-1.5. In this setting, the next token is selected deterministically by maximizing the conditional probability at each step, thereby eliminating randomness from the generation process.

As shown in Table 7, our method remains consistently strong in the fully deterministic setting. In particular, it achieves the best performance on the challenging POPE-Adversarial split and simultaneously yields the lowest $CHAIR_S$ and $CHAIR_I$,

| Method | Random | | Popular | | Adversarial | | CHAIR | |
|---|---|---|---|---|---|---|---|---|
| | Acc ↑ | F1 ↑ | Acc ↑ | F1 ↑ | Acc ↑ | F1 ↑ | CHAIR$_S$ ↓ | CHAIR$_I$ ↓ |
| Greedy | 89.33 | **89.28** | 85.93 | 86.33 | 79.10 | 80.95 | 50.9 | 15.40 |
| VCD | 87.97 | 87.84 | 85.33 | 85.67 | 78.43 | 80.26 | 54.6 | 16.60 |
| OPERA | 89.27 | 88.72 | 86.80 | 86.59 | 81.13 | 81.87 | 49.0 | 13.52 |
| SID | 89.33 | 89.10 | 86.47 | 86.25 | 81.77 | 82.31 | 48.0 | 13.60 |
| **Fox (Ours)** | **89.43** | 88.81 | **87.07** | **86.60** | **82.43** | **82.64** | **42.6** | **12.16** |

*Table 7.* **Greedy decoding results on POPE and CHAIR.** We report split-wise POPE Accuracy and F1 on Random/Popular/Adversarial, together with CHAIR$_S$ and CHAIR$_I$. Since greedy decoding is used as a diagnostic control to eliminate sampling stochasticity, we omit averaged scores and focus on split-level robustness across difficulty regimes.

indicating fewer hallucinated objects at both the sentence and instance levels. These improvements persist without any sampling variance, suggesting that the gains are not attributable to temperature tuning or favorable sampling randomness, but stem from the proposed structural intervention.

These results provide strong evidence that our improvements stem from *structural intervention on high-risk mediators*, rather than from stochastic decoding effects. In particular, even in the absence of sampling diversity, our method effectively suppresses hallucination while maintaining semantic coverage, as reflected by improvements on both POPE and CHAIR metrics.

### C.8 Mitigating Generation Degradation via Conflict-Gated Cooperation

As demonstrated in our previous analysis, the structural intervention on risky mediators effectively severs the shortcut path $\mathbf{X}_{sys} \to H_R \to Y_t$, thereby promoting visual grounding. However, as illustrated in Figure 19, this "de-priorization" process can lead to unintended consequences. While the interventional branch $P_{do}$ achieves high factual accuracy by strictly relying on stable visual evidence, it simultaneously compresses the model's linguistic expressive capacity, occasionally resulting in repetitive patterns or a lack of semantic richness, a common challenge for attention-based suppression methods.

**Rationale for JSD-based Cooperative Decoding.** To address this, we employ a Conflict-Gated Cooperative Decoding strategy (Insight III) based on Jensen-Shannon Divergence ($d_t$). The rationale for using JSD as the gating mechanism is twofold:

- **Real-time Conflict Detection:** $d_t$ serves as a sensitive probe to detect when the interventional branch significantly deviates from the observational manifold. High divergence suggests that the model is at a critical decision point where the intervention might be over-suppressing necessary linguistic context.
- **Adaptive Manifold Re-injection:** By utilizing JSD to regulate the fusion, we effectively re-inject the "high-precision anchor" ($P_{do}$) into the "high-fluency manifold" ($P_{obs}$). As shown in Figure 19, compared to pure attention intervention which suffers from context loss and repetition (red text), our Entropy-Guided Causal Decoding (Fox) achieves a superior balance, maintaining factual accuracy (green text) while preserving the natural flow and diversity of the generated response.

### C.9 Additional Performance

We present additional qualitative examples to showcase the practical performance of Fox in reducing hallucinations across different LVLM backbones. As shown in Fig. 20, Fox effectively suppresses prior-driven hallucinated attributes and objects on LLAVA-1.5 while preserving visually grounded details. Fig. 21 further demonstrates that the proposed intervention generalizes to INSTRUCTBLIP. Finally, Fig. 22 reports additional cases on INSTRUCTBLIP and SHIKRA, where Fox consistently improves visual grounding under diverse scenes and object configurations. These examples complement the quantitative results in the main paper by providing intuitive evidence of cross-backbone robustness.

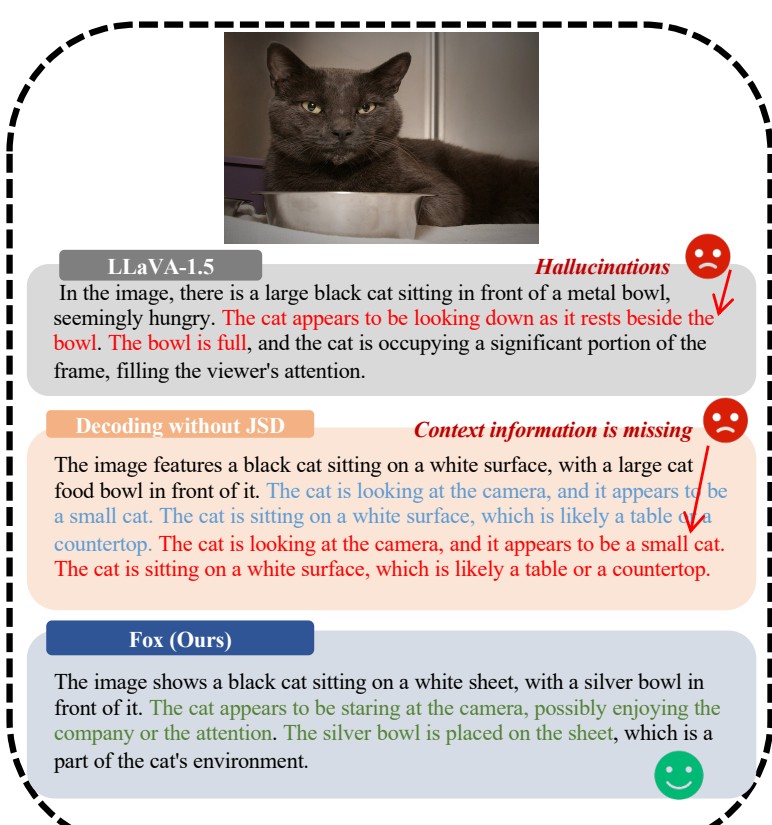

*Figure 19.* **The role of JSD-based conflict gating.** Without JSD gating, always applying the intervention leads to severe generation degradation due to excessive context suppression. Conversely, an overly large JSD threshold biases the decoding toward an overly conservative regime, reducing semantic coverage. A moderate JSD threshold enables adaptive cooperation between the interventional and observational branches, achieving a balanced trade-off between factual reliability and generation quality.

**LLaVA-1.5 (Sampling)**

The image features a woman in a red shirt sitting on a wooden bench in a park. She is positioned in the middle of the bench, with her legs crossed. The woman is wearing glasses, and she has a hat on, giving her a casual and relaxed appearance. In the scene, there are some additional items around the bench. There are three potted plants nearby, two on the left and one on the right, adding a touch of nature to the surroundings. A handbag is placed on the bench near the woman, possibly belonging to her. Additionally, there are a few cars and a truck parked in the background, giving a hint of the park's location or parking area.

**Fox (Ours)**

The image features a woman sitting on a wooden bench in a park. She is wearing a red shirt and a white hat. The bench is located in a shaded area with a few trees surrounding it. The woman is posing for the camera with a smile on her face, and she is resting her hand on the bench. The scene is a peaceful, outdoor setting with a relaxed atmosphere.

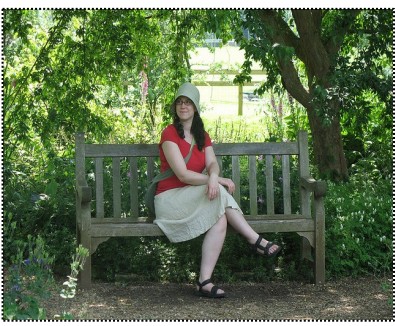

**LLaVA-1.5 (Sampling)**

The image features a small child sitting outside wearing a camouflage sleeping outfit. The child is engaged in holding a small electronic device, possibly a smartphone, while sitting on the floor.Around the child, there is a door to the left and two more doors visible at the back. In the background, there are also two cups, one on the far right side and the other closer to the left side of the image. Additionally, there is a handbag located on the left side of the scene.

**Fox (Ours)**

The image depicts a young boy in camouflage pajamas sitting on a rug. He is holding a cell phone in his hand, possibly playing a game or interacting with the device. The boy is seated on the floor in front of a doorway, with a tiled floor visible in the background.

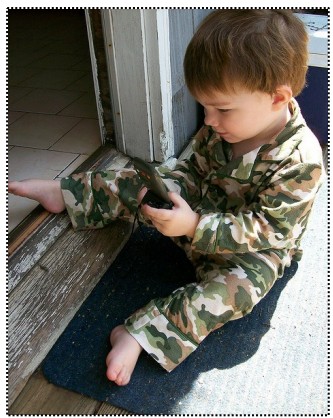

**LLaVA-1.5 (Sampling)**

The image features a yellow and blue train car with various types of graffiti covering its exterior, including the train's door. There are multiple windows present, both on the train car and on the doors, giving passengers an opportunity to see out. Additionally, there are two people visible in the scene, with one person standing closer to the train and another person further away from it. The graffiti on the train car is quite vibrant and eye-catching, which makes it stand out as a unique and artistic form of expression.

**Fox (Ours)**

The image features a train with graffiti on it, as well as a yellow and blue color scheme. The train is parked at a station, possibly in a busy city. The graffiti covers the train's exterior, adding a unique and artistic touch to the public transportation.

*Figure 20.* Fox's performance on reducing hallucinations of LLaVA-1.5.

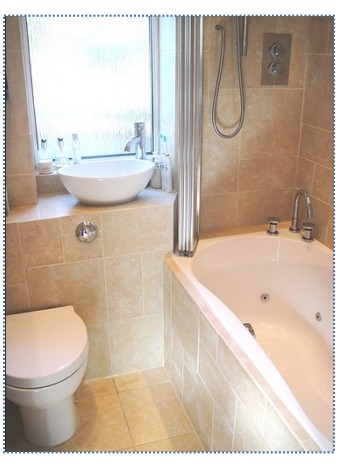

**Shikra (Sampling)**

The image features a modern bathroom with a white toilet, sink, and bathtub. The sink is situated in the corner of the bathroom, while the toilet and bathtub are placed adjacent to each other. A glass shower is visible in the scene, adding to the sense of sophistication in the design.Various toiletries, including multiple bottles, are present near the sink, providing a functional space for bathroom essentials. There is also a cup placed near the sink area. The overall appearance of the bathroom suggests a clean and well-maintained space for users to enjoy.

**Fox (Ours)**

The image shows a bathroom with a toilet, sink, and bathtub. The sink is located in the left corner of the bathroom, while the toilet is positioned on the right side of the room. The bathtub is situated in the middle of the bathroom, with its end section partially visible. In addition to the sink and toilet, there are several bottles placed around the bathroom, some near the sink and others near the bathtub. The bottles seem to be various bathroom items, possibly for personal care and cleaning purposes.

**Shikra (Sampling)**

The image features a small black bicycle, which is leaning against a wooden post in the center of a gravel area. The bike appears to be quite nice-looking and well-maintained. There is also a second, smaller bicycle situated nearby, although it is not leaning against a post. Two people can be seen in the background, possibly observing or attending to the bicycles. Additionally, a couple of benches are visible in the scene, one located in the middle of the area and the other one farther behind.

**Fox (Ours)**

The image features a green bicycle parked in a gravel lot, situated in front of a wooden fence. The bicycle is leaning against a wooden pole, and the fence appears to be made of wood. There are also some bushes nearby, adding a touch of greenery to the scene.

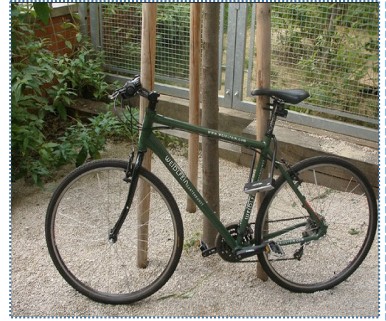

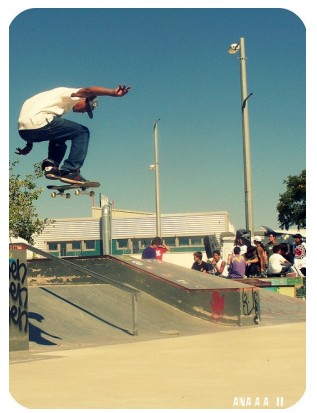

**Shikra (Sampling)**

The image captures a lively scene at a skate park where a man is performing an impressive trick on a skateboard, jumping over a set of stairs. He is the center of attention, with several other people watching his skills from various positions around the park. In total, there are 14 other people, some of which are standing close to the skateboarder, while others are at a bit of a distance, admiring the display. In addition to the skateboarder, there is a bicycle visible in the background on the right side of the scene. The onlookers are engaged and enjoying the spectacle, creating a vibrant atmosphere at the skate park.

**Fox (Ours)**

The image features a male skateboarder in the midst of performing a trick, captured mid-air on his skateboard. He is skillfully jumping over a set of stairs, showcasing his talent and control. There are several onlookers in the scene, watching the skateboarder with interest. They are scattered around the area, with some standing close to the staircase and others further away. The onlookers are watching the skateboarder as he completes his daring jump, likely impressed by his performance.

*Figure 21.* Fox's performance on reducing hallucinations of InstructBLIP.

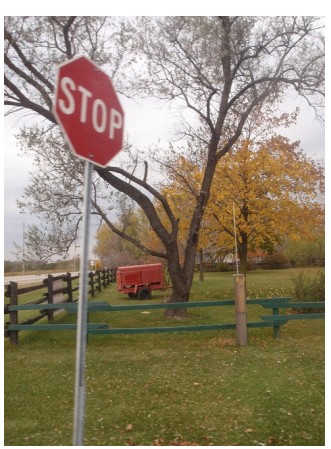

**InstructBLIP (Sampling)**

The image features a small town setting with a road leading to a green field. In the center of the scene, there is a stop sign placed at a corner. A few trees are visible throughout the field, and one is specifically in the background. A red barrel is visible near the stop sign, possibly signifying a recycling bin or some sort of container. A truck is parked near the roadside, and two parked cars can be seen in the field, providing some sense of background context. Overall, the scene suggests a small community with a focus on environmental sustainability.

**Fox (Ours)**

The image shows a stop sign situated on a grassy area with a fence and a tree in the background. The stop sign is located in the center of the image, with a red color and a white border. The stop sign is placed on a grassy area, with a fence and a tree nearby. The fence is positioned on the right side of the image, with a tree on the left side. The grassy area is surrounded by a fence, which is visible on the left side of the image.

**InstructBLIP (Sampling)**

The image features a group of people playing a frisbee game in an open grassy field. The people are both men and women, with some holding frisbees, while others appear to be watching or cheering on the game. There are three frisbees visible in the scene, including one being thrown by a person, another being caught by someone, and a third resting on the grass. In total, there are 10 people visible, five of whom are holding frisbees. Some are dressed in red, indicating they might belong to the same team, while others appear to be in black and white or other colors. There are several cups placed around the area, suggesting there might be a drink available for the participants.

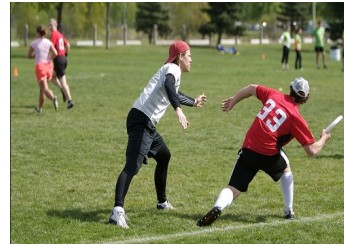

**Fox (Ours)**

The image shows a group of people playing a game of ultimate frisbee in a park. The scene is lively, with a variety of people playing, running, and throwing the frisbee. There are several people in the scene, with some standing around and others actively participating in the game. In total, there are at least six people in the image, with some being more active in the game and others standing around. The game is taking place in a spacious park, with a large open field and a few trees in the background.

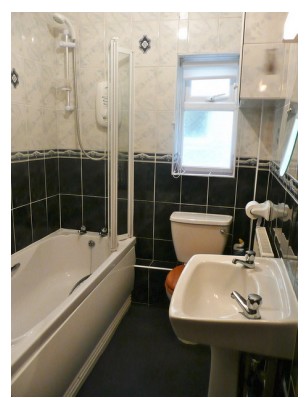

**InstructBLIP (Sampling)**

The image features a well-appointed bathroom with a bathtub, sink, and toilet. The sink is placed close to the door of the bathroom, while the bathtub is in the center of the space, with a window providing natural light. The toilet is further into the bathroom, away from the sink and tub. A mirror is placed on the wall adjacent to the sink, allowing for a full reflection of oneself while performing tasks. Additionally, a bottle can be seen near the sink, presumably filled with toiletries.

**Fox (Ours)**

The image shows a bathroom with a sink, toilet, and a bath tub. The sink is located on the right side of the room, and the toilet is situated on the left side. The bath tub is located in the center of the room, providing a spacious and functional area for bathing. The room has a black and white color scheme, with the black tiles on the walls and the white sink and bathtub. The sink is slightly larger than the toilet, creating a unique and functional layout. The window is also visible in the room, providing natural lighting and a view of the outside.

*Figure 22.* Fox's performance on reducing hallucinations of Shikra.

