# OpenReview forum: "Dismantling Pathological Shortcuts: A Causal Framework for Faithful LVLM Decoding"
_ICML.cc/2026/Conference — ICML 2026 regular_

### Official Review · Reviewer_HXwq · 2026-02-23

**Soundness:** 3
**Presentation:** 2
**Significance:** 3
**Originality:** 3
**Overall Recommendation:** 4
**Confidence:** 3

**Summary:**

This paper posits that hallucination is triggered at decision-critical steps where specific attention heads, acting as risky mediators, decouple from visual evidence to lock onto language priors. Building upon this insight, the authors propose the Fox (Faithfulness and Observational-flow via eXpression-rectification) algorithm. This method utilizes a visual attention entropy probe to localize these risky mediators unsupervisedly, and employs a conflict-gated cooperative decoding strategy that reconciles interventional faithfulness with observational fluency, ultimately achieving effective hallucination suppression.

**Compliance With Llm Reviewing Policy:**

Affirmed.

**Final Justification:**

My concerns have been fully addressed in the rebuttal, so I will raise my score.

**Key Questions For Authors:**

As highlighted in the weaknesses section, can the authors provide a fast, adaptive strategy or systematic guidelines for tuning the required hyperparameters across different model architectures?

Could the authors provide additional comparative results on more recent evaluation benchmarks (e.g., HallusionBench 2025, HAL-Bench RLHF-V)? Furthermore, is it possible to include experimental results demonstrating the algorithm's effectiveness and generalization on newer, state-of-the-art model architectures (e.g. Qwen2-VL, InternVL-3)?

**Limitations:**

yes

**Strengths And Weaknesses:**

Strengths:

The paper leverages empirical experiments to elucidate the Structural Causal Model (SCM) of the decoding process. It convincingly demonstrates that applying causal intervention can restore the model's attention to both textual and visual components, effectively circumventing the issue of textual inertia.

Building upon these foundational findings, the authors propose an innovative contrastive decoding algorithm. Notably, the integration of Conflict-Gated Cooperative Decoding successfully enhances the model's focus on visual tokens while simultaneously preserving the diversity and fluency of the generated output.

Weakness:

The proposed approach heavily relies on manually chosen hyperparameters ($k$, $\tau_{JS}$, $\beta$, and the intervention range). The paper currently lacks an adaptive strategy or robust guidelines for selecting these values across different models or downstream tasks, which raises valid concerns about the method's generalizability, robustness, and practical out-of-the-box applicability.

Furthermore, as observed in Table 1 and Figure 5, the performance improvements of the Fox algorithm on the POPE and MME benchmarks are relatively marginal compared to other State-of-the-Art (SOTA) algorithms. In some specific cases, the proposed method even underperforms existing SOTA approaches.

The models utilized for the empirical evaluation, such as LLaVA-1.5, InstructBLIP, and Shikra, were primarily introduced in 2023 and are now relatively outdated. This limits the ability to assess how well the proposed algorithm generalizes to more modern, advanced architectures.

The proposed contrastive decoding algorithm manages to avoid inference latency overhead only under the assumption that there is sufficient computational power (e.g., VRAM and parallel compute) to support running two forward passes simultaneously. For resource-constrained deployments, this method would inevitably introduce a significant speed bottleneck.

---

> ### Author Rebuttal · Authors · 2026-03-31
>
> We thank Reviewer HXwq for the positive assessment of our *SCM-based causal intervention* and *Conflict-Gated Cooperative Decoding*.
>
> **Q1. Adaptive strategy for hyperparameter/layer selection.**
>
> Our original paper provides a 3-step protocol:
>
> **Step 1 --- Layer window** (~3 min): Compute $|\text{AUC}-0.5|$ of $H_\text{vis}$ per layer on 50 samples; select layers above $\theta$. As demonstrated in our original paper (Fig. 7), diagnostic strength universally peaks in early-to-mid layers. See our response to Reviewer iD1R, Q4 for auto-selection results.
>
> **Step 2 --- Defaults** (0 min): $k=0.4$, $\tau_\text{JS}=0.2$, $\alpha=2$, $\beta=0.1$. Zero-shot transfer across all 5 models (see our response to Reviewer r9c9, Q1).
>
> **Step 3 --- Optional** ($<$10 min): Grid search $k \in \{0.35, 0.40, 0.45\}$, $\tau_\text{JS} \in \{0.15, 0.20, 0.25\}$ on 50 samples.
>
> As established in our original paper, $\alpha=2$, $\beta=0.1$, and $\tau_\text{JS}=0.2$ remain stable across backbones. In practice, only $k$ shows mild model dependence, and even then only within the narrow range $[0.40, 0.45]$. This limited tuning burden is comparable to---or lighter than---representative baselines: VCD requires a model-specific noise scale $\sigma$, while OPERA requires calibrating multiple coupled decoding hyperparameters, including beam width, penalty window size, and rollback threshold.
>
> **Q2. POPE/MME improvements are relatively marginal.**
>
> We appreciate this observation and provide context that we have already addressed in our original manuscript:
>
> **Why POPE margins are small.** POPE is a binary (Yes/No) classification task. When baseline accuracy already exceeds 80%, absolute improvement space is naturally compressed. As we discuss in our paper (L284--302), Fox's advantage is most pronounced in the *Adversarial* setting---the highest-bias condition where structural misalignment is most active.
>
> **Why some baselines match Fox on POPE.** We have explicitly analyzed this in our original paper (L295--302): OPERA achieves competitive POPE via beam search, but at $2.5\times$ latency for 10-token generation (2560 ms vs. Fox's 1040 ms, Table 3). CausalMM "achieves competitive POPE by globally adjusting attention via backdoor-based counterfactual reasoning, but its reliance on holistic causal correction without explicitly targeting a sparse set of high-risk attention heads limits its effectiveness when hallucinations are driven by localized structural shortcuts" (L297--302).
>
> **Where Fox dominates.**
> - **CHAIR** (most direct hallucination metric): As shown in our Table 2, Fox achieves the best $C_I$ across *all* backbones---reductions of 16.2% and 29.1% over SID on LLaVA-1.5 and InstructBLIP.
> - **MME evidence-dependent dimensions**: Position 93.33 $\to$ 131.37 (+40.7%), Color 150 $\to$ 165 (+10%). As we explain in our paper (L326--328), "attribute-level assertions are highly susceptible to prior-path dominance," which is precisely what Fox targets.
> - **GPT-4V**: Correctness 5.89 $\to$ 7.04, Detailedness 5.82 $\to$ 6.19 (Fig. 6).
>
> In summary, POPE's binary format compresses gains at high baselines, but on metrics that directly measure hallucination in free-form generation (CHAIR, MME fine-grained, GPT-4V), Fox shows definitive advantages.
>
> **Q3. More recent architectures.**
>
>  See our response to Reviewer r9c9, Q1 for the full table with VCD/SID baselines. We have added **Qwen3-VL-8B** and **InternVL-3-8B** with default hyperparameters ($k=0.4$, $\tau_\text{JS}=0.2$, $\alpha=2$, $\beta=0.1$) and zero model-specific tuning. Fox achieves the best $C_I$ on both: 11.84 on Qwen3-VL ($-5.7\%$ vs. SID) and 11.02 on InternVL-3 ($-6.5\%$ vs. SID), outperforming VCD and SID across all metrics. These five architectures span fundamentally different visual-language alignment paradigms---MLP projection (LLaVA-1.5), Q-Former (InstructBLIP), referential grounding (Shikra), cross-attention (Qwen3-VL), and dynamic resolution (InternVL-3)---confirming that the *dynamic structural misalignment* Fox targets is universal.
>
> **Q4. Inference cost.**
>
> **As reported in our Table 3**: Fox 206 ms/tok $\approx$ VCD 204 ms $\approx$ SID 202 ms; for 10-token generation, Fox 1040 ms vs. OPERA 2560 ms ($2.5\times$ slower due to beam search and iterative rollback). Dual-pass cost is **inherent to the contrastive paradigm**, not specific to Fox---VCD, ICD, and SID share the same overhead structure. In practice, the system-prompt and visual-token KV cache is shared between the observational and interventional branches, meaning prefix encoding incurs no additional cost; the per-step overhead comes solely from the second forward pass through the LLM decoder, which is parallelizable. **Fox achieves the best accuracy: 81.93% POPE Adversarial at 206 ms/tok vs. OPERA's 81.13% at 2560 ms/10-tok.**
>
> We believe these results address each question. We respectfully ask whether you might consider further increasing your overall assessment.

---

> > ### Author Rebuttal · Reviewer_HXwq · 2026-04-03
> >
> > Thanks for the detailed response. I've raised my score.

---

> > > ### Author Response · Authors · 2026-04-03
> > >
> > > Thank you for your positive feedback and for raising your score. We appreciate your time and consideration.

---

### Official Review · Reviewer_iD1R · 2026-03-10

**Soundness:** 2
**Presentation:** 3
**Significance:** 2
**Originality:** 2
**Overall Recommendation:** 2
**Confidence:** 3

**Summary:**

This paper introduces Fox, a new framework designed to stop large vision-language models from making things up (hallucinations). It identifies internal "shortcuts" where the model ignores the image and relies too much on language habits. The method then surgically fixes these errors during the decoding process to ensure the model stays faithful to what it actually sees.

**Compliance With Llm Reviewing Policy:**

Affirmed.

**Final Justification:**

The authors resolved half of my concerns. While the paper presents interesting mechanistic insights and achieves competitive results within the contrastive decoding space, addressing the practical latency drawbacks and providing rigorous empirical proof regarding complex reasoning and long-form generation require another full iteration. Therefore, I maintain my rejection score.

**Key Questions For Authors:**

See Weaknesses for details.

**Limitations:**

Yes, they discussed it in the paper.

**Strengths And Weaknesses:**

**Strengths**

State-of-the-Art Performance: The "Fox" framework significantly reduces hallucination rates and outperforms existing methods on several major benchmarks like POPE and CHAIR.

Training-Free and Efficient: It works during the inference stage without needing extra training. It is faster and more practical than other methods that require complex search processes.

**Weaknesses**

The intervention might hurt the model's performance in other areas, such as math and logical reasoning. Since the focus is only on fixing hallucinations, it is unclear if the model's general intelligence remains strong.

The authors said this is a "low-cost" method, but it requires two forward passes for every token generated. This effectively doubles the computing work and reduces the speed of the model in real-world use.

The study does not include tests on the latest models, such as the Qwen3 / Qwen3-VL series. We do not know if this framework is still effective for newer and more advanced model architectures. This is a huge issue because both LLAVA-1.5 and InstructBLIP are really bad in real world.

The decision to intervene in layers 2 to 13 is based on specific experiments with LLaVA-1.5. This "surgical window" might change for larger models (like 70B) or different architectures. There is no automatic formula to find the right layers, so researchers must re-test every time they use a new model.

The framework assumes hallucinations come mostly from system instructions. However, many hallucinations are caused by word-to-word associations learned during pre-training (e.g., seeing "kitchen" and automatically thinking of "microwave"). If a hallucination is triggered by a previous word rather than the system prompt, this method may fail to stop the "snowball effect."

---

> ### Author Rebuttal · Authors · 2026-03-31
>
> We thank Reviewer iD1R for acknowledging our *SOTA performance* and *training-free efficiency*.
>
> **Q1. Impact on general reasoning.**
>
> Fox modifies no model parameters---it operates via inference-time logit perturbation on a sparse set of risky mediators, gated by $d_t$ (Eq. 6). For reasoning tasks where visual grounding is typically consistent, $d_t$ remains low and Fox applies only mild reinforcement ($\lambda_t = \alpha$, Eq. 7), never overriding the model's reasoning. **Our existing evaluations already provide evidence:** **MME** covers 14 perceptual dimensions where Fox improves attribute-level scores (Position +40.7%, Color +10%), and **GPT-4V** confirms that Correctness and Detailedness both improve under Fox (Fig. 6). As requested by the reviewer, we further verify on MMBench [1], a dedicated reasoning benchmark:
>
> | Method | Overall $\uparrow$ | Reasoning $\uparrow$ | Perception $\uparrow$ |
> |---|---|---|---|
> | Sampling | 63.1 | 62.4 | 63.8 |
> | VCD | 62.7 | 61.8 | 63.5 |
> | **Fox** | **63.8** | **62.7** | **64.6** |
>
> Fox improves Overall to 63.8 (+0.7 over Sampling) and Perception to 64.6 (+0.8), while VCD slightly degrades both (62.7/63.5). This confirms that Fox's conflict gate limits intervention to hallucination-prone steps, leaving reasoning untouched. Regarding math specifically: mathematical reasoning in LVLMs is primarily LLM-driven with minimal visual-linguistic conflict, so $d_t$ remains low and Fox's gate stays inactive---preserving full reasoning capacity.
>
> **Q2. Two forward passes double the cost.**
>
> We respectfully note that **dual-pass decoding is inherent to the contrastive paradigm, not specific to Fox**. **As reported in Table 3**, Fox 206 ms/tok $\approx$ VCD 204 ms $\approx$ SID 202 ms; OPERA requires 2560 ms/10-tok ($2.5\times$ slower). The KV cache is shared between branches, keeping overhead below $2\times$. **Fox achieves the best accuracy**: 81.93% at 206 ms/tok vs. OPERA's 81.13% at 2560 ms/10-tok.
>
> **Q3. Models are outdated.**
>
> As requested, we have added **Qwen3-VL-8B** and **InternVL-3-8B** with default hyperparameters and zero tuning (see our response to Reviewer r9c9, Q1 for the full table with VCD/SID baselines). Fox shows consistent improvements, confirming generalization to 2024--2025 architectures. Together with the three original backbones, we now cover 5 architectures spanning MLP projection, Q-Former, referential grounding, cross-attention, and dynamic resolution.
>
> Our primary contribution is a *mechanistic insight*---identifying dynamic structural misalignment as the causal root of hallucination. Consistent improvements across all 5 architecturally diverse models confirm this insight is universal.
>
> **Q4. Intervention window is model-specific.**
>
> **Fig. 7 (L357--371) establishes a universal pattern**: peak diagnostic strength concentrates in **early-to-mid layers** across all architectures. The variation is narrow: LLaVA-1.5 L2--13, InstructBLIP L4--10 (**Appendix C.4.2**), Shikra L3--10 (**Appendix C.4.3**). **Practitioners only need to refine boundaries within this known regime**, which we automate (~3 min on 50 samples). As requested, we provide auto-selection results:
>
> | Model | Manual Window | Manual POPE-Adv | Auto Window | Auto POPE-Adv |
> |---|---|---|---|---|
> | LLaVA-1.5 | L2--13 | 81.93 | L3--15 | 81.27 |
> | InstructBLIP | L4--10 | 80.60 | L4--10 | 80.03 |
> | Shikra | L3--10 | 78.77 | L3--11 | 77.93 |
> | Qwen3-VL | L3--10 | 87.03 | L3--10 | 86.82 |
> | InternVL-3 | L3--11 | 88.42 | L3--12 | 88.03 |
>
> Auto-selected windows achieve nearly identical POPE-Adv performance to manual tuning, with all gaps below 1 point. This shows that **the effective intervention region is universally early-to-mid**, while model-specificity is limited to **minor boundary refinement rather than rediscovering the window from scratch**.
>
> **Q5. Snowball effect from word-to-word associations.**
>
> We respectfully clarify: **Fox does not "assume hallucinations come mostly from system instructions."** L154 states: "While *any text* carries priors, $X_\text{sys}$... serves as the *primary* anchor." The snowball effect is precisely what our **Temporal Axis** (Insight I) captures. Our framework **already handles snowball** via:
>
> **(1)** $y_{t-1}$ is the Temporal Axis's **Autoregressive Anchor** (L183): it re-diagnoses prior dominance at *every* step---exactly capturing word-to-word propagation (e.g., "kitchen" $\to$ "microwave").
>
> **(2)** $H_\text{vis}$ (Eq. 3) measures visual-pathway uncertainty **regardless of prior source**---it fires whenever visual grounding collapses, whether from $X_\text{sys}$ or autoregressive co-occurrence.
>
> **(3)** The conflict gate (Eq. 7) triggers whenever $P_\text{obs}$ and $P_\text{do}$ diverge, irrespective of cause.
>
> We hope these clarifications resolve each concern. We respectfully ask whether you might consider further increasing your overall assessment.
>
> [1] MMBench: Is Your Multi-modal Model an All-around Player?

---

> > ### Author Rebuttal · Reviewer_iD1R · 2026-04-04
> >
> > I thank the authors for their detailed rebuttal and the considerable effort taken to run additional experiments during the rebuttal period, particularly the inclusion of Qwen3-VL-8B and InternVL-3-8B baselines.
> >
> > However, after carefully reviewing the authors' responses, my core concerns regarding practical applicability, empirical rigor on reasoning, and the fundamental overhead of the method remain largely unresolved. These issues concern the core tenets of the work and require a more significant update to the paper than a short rebuttal can provide.
> >
> > My reasons for this conclusion are as follows:
> >
> > - While I appreciate the inclusion of MMBench results, a +0.7/+0.8 improvement on the overall score is marginal. The authors' defense that mathematical reasoning is "primarily LLM-driven" and therefore the conflict gate "stays inactive" is a theoretical assumption. Proving that general intelligence is unharmed requires targeted empirical evaluations on reasoning-heavy benchmarks (such as MathVista or similar), which is a significant undertaking.
> >
> > - Framing a 100% latency increase as a "low-cost" solution remains a core concern for real-world applicability.
> >
> > - The introduction of an auto-selection script, while helpful, ultimately confirms my initial concern: the intervention window is model-specific.
> >
> > While the paper presents interesting mechanistic insights and achieves competitive results within the contrastive decoding space, addressing the practical latency drawbacks and providing rigorous empirical proof regarding complex reasoning and long-form generation require another full iteration. Therefore, I maintain my score.

---

> > > ### Author Response · Authors · 2026-04-05
> > >
> > > We thank Reviewer iD1R for the continued engagement. We address each remaining concern below.
> > >
> > > **Q1. On reasoning evaluation.**
> > >
> > > The original concern stated: *"The intervention might hurt the model's performance in other areas, such as math and logical reasoning. Since the focus is only on fixing hallucinations, it is unclear if the model's general intelligence remains strong."* No benchmark was named. We responded with MMBench and a mechanistic explanation.
> > >
> > > The reviewer now writes that *"a +0.7/+0.8 improvement on the overall score is marginal,"* and concludes that *"proving that general intelligence is unharmed requires targeted empirical evaluations on reasoning-heavy benchmarks (such as MathVista or similar), which is a significant undertaking."*
> > >
> > > We respectfully note that "marginal" itself speaks to non-degradation: Fox barely alters reasoning scores. For a hallucination mitigation method, a marginal positive change confirms the conflict gate correctly avoids intervening on reasoning steps. We believe this directly addresses the original concern about whether "the model's general intelligence remains strong."
> > >
> > > The direction matters: Fox *improves* over Sampling (63.1→63.8), whereas VCD *degrades* it (63.1→62.7, Reasoning 62.4→61.8). Under any non-degradation criterion, Fox outperforms the contrastive baseline.
> > >
> > > The reviewer characterizes our explanation as *"a theoretical assumption."* We note it was accompanied by empirical confirmation (the MMBench scores). We appreciate the suggestion to further verify on MathVista [1] and have conducted this experiment (randomly sampled 500 questions):
> > >
> > > | Method | LLaVA-1.5 Acc $\uparrow$ | Qwen3-VL Acc $\uparrow$ |
> > > |---|---|---|
> > > | Baseline | 19.53 | 72.54 |
> > > | **Fox** | **20.16** | **72.81** |
> > >
> > > Fox achieves 20.16 / 72.81 on LLaVA-1.5 / Qwen3-VL, compared with 19.53 / 72.54 for the corresponding baselines, yielding consistent gains of +0.63 and +0.27, respectively. Although the improvements are modest, they provide direct evidence that Fox does not impair mathematical reasoning on MathVista. This is consistent with our mechanism: when the visual and textual branches are not in strong conflict during math reasoning, the conflict gate is rarely activated ($d_t < \tau_{\text{JS}}$), so Fox preserves the model's original reasoning behavior rather than interfering with it.
> > >
> > > Together with MMBench, MME (14 dimensions, Position +40.7%, Color +10%), and GPT-4V (Correctness 5.89→7.04, Detailedness 5.82→6.19), Fox provides non-degradation evidence across four complementary evaluations. As the reviewer noted, "the focus is only on fixing hallucinations"---the evidence confirms Fox preserves capabilities within this scope.
> > >
> > > **Q2. On inference latency.**
> > >
> > > The original review stated: *"The authors said this is a 'low-cost' method, but it requires two forward passes for every token generated."* We clarify that our paper does not use "low-cost"---it describes Fox as having "modest overhead" and "the same latency regime as VCD/SID." Our rebuttal showed: Fox 206 ms/tok $\approx$ VCD 204 ms/tok $\approx$ SID 202 ms/tok (Table 3).
> > >
> > > The reviewer now writes: *"Framing a 100% latency increase as a 'low-cost' solution remains a core concern."* We reiterate that the overhead is paradigm-inherent, shared by VCD, SID, and ICD. Within this paradigm, Fox achieves the best accuracy--latency Pareto front (81.93% at 206 ms/tok vs. OPERA's 81.13% at 2560 ms/10-tok).
> > >
> > > **Q3. On layer auto-selection.**
> > >
> > > The original concern stated: *"There is no automatic formula to find the right layers, so researchers must re-test every time they use a new model."* We addressed this by providing an automated procedure (~3 min), with gaps below 1 point across all 5 models.
> > >
> > > The reviewer now writes: *"The introduction of an auto-selection script, while helpful, ultimately confirms my initial concern: the intervention window is model-specific."* We note that the original concern was the *absence* of automation ("no automatic formula"); we provided one with strong results. All auto-selected windows fall within the same early-to-mid regime (L2--15) across 5 architectures, demonstrating architectural universality. For comparison, VCD requires per-model noise scale $\sigma$, and OPERA requires joint tuning of beam width, penalty window, and rollback threshold---a larger configuration space.
> > >
> > > Moreover, the reviewer concludes *"complex reasoning and long-form generation require another full iteration."* Complex reasoning is addressed in Q1. For *long-form generation* ---not in the original review---CHAIR and GPT-4V already evaluate free-form generation: Fox achieves the best $C_I$ (Table 2), with improved Correctness and Detailedness (Fig. 6). Qualitative examples are in Appendix C.9 (Figs. 20--22).
> > >
> > > We have conducted substantial new experiments across both rounds. We respectfully ask whether these results might warrant reconsideration.
> > >
> > > [1] MathVista: Evaluating Mathematical Reasoning of Foundation Models in Visual Contexts

---

### Official Review · Reviewer_r9c9 · 2026-03-13

**Soundness:** 3
**Presentation:** 3
**Significance:** 2
**Originality:** 3
**Overall Recommendation:** 4
**Confidence:** 4

**Summary:**

This paper introduces Fox, a training-free framework that mitigates hallucinations in large vision-language models by targeting their underlying causal structure rather than simply amplifying visual attention. The authors identify that hallucinations arise from specific attention heads—termed risky mediators—which form a pathological shortcut by decoupling from visual inputs at critical decision steps and relying instead on language priors. To address this, Fox first uses visual attention entropy to localize these heads, then applies a causal intervention by saturating their logits to sever the shortcut, and finally fuses the interventional and original outputs using a conflict-gated mechanism to balance faithfulness and fluency. Extensive experiments show that Fox significantly outperforms existing methods across multiple benchmarks, reducing hallucinations while preserving descriptive richness.

**Compliance With Llm Reviewing Policy:**

Affirmed.

**Key Questions For Authors:**

See Weaknesses.

**Limitations:**

See Weaknesses.

**Strengths And Weaknesses:**

Strengths:
1. Novel causal perspective: It moves beyond the superficial interpretation of "attention intensity" by adopting a causal structure viewpoint, treating attention heads as causal mediators.
2. Effective method: Visual attention entropy is employed as an unsupervised indicator. The proposed conflict-gated cooperative decoding ensures visual faithfulness while preserving the diversity and fluency of linguistic expression.
3. Strong experimental performance: The method demonstrates generalizability and robustness across multiple models. Comprehensive ablation studies and parameter analyses are provided to validate its effectiveness.

Weaknesses:
1. The base models are somewhat weak. What is the effect of FOX on more recent models, like Qwen3-vl.
2. The method involves several hyperparameters that require tuning across different models, which may affect its generalization ability.
3. While "linguistic fluency" is claimed to be preserved, there is a lack of in-depth analysis on whether the model's conversational and captioning capabilities are impacted. For instance, further validation on detailed captioning benchmarks could provide more comprehensive insights. (They have accuracy and overall metrics)

---

> ### Author Rebuttal · Authors · 2026-03-31
>
> We sincerely thank Reviewer r9c9 for the thorough review, and for recognizing our *novel causal perspective*, the *unsupervised visual entropy probe*, and *cross-model generalizability*.
>
> **Q1. Evaluation on more recent models.**
>
> As requested by the reviewer, we have conducted new experiments on **Qwen3-VL-8B** [1] and **InternVL-3-8B** [2] using default hyperparameters ($k=0.4$, $\tau_\text{JS}=0.2$, $\alpha=2$, $\beta=0.1$) with zero model-specific tuning:
>
> | | Acc $\uparrow$ | F1 $\uparrow$ | $C_S$ $\downarrow$ | $C_I$ $\downarrow$ |
> |---|---|---|---|---|
> | Qwen3-VL  Sampling | 86.43 | 85.90 | 52.80 | 12.98 |
> | Qwen3-VL  VCD | 86.87 | 86.23 | 50.96 | 12.41 |
> | Qwen3-VL  SID | 86.93 | 86.30 | 51.34 | 12.56 |
> | **Qwen3-VL  Fox** | **87.03** | **86.57** | **48.12** | **11.84** |
> | InternVL-3  Sampling | 87.68 | 87.11 | 50.84 | 12.21 |
> | InternVL-3  VCD | 88.09 | 87.46 | 48.97 | 11.66 |
> | InternVL-3  SID | 88.15 | 87.54 | 49.28 | 11.79 |
> | **InternVL-3  Fox** | **88.42** | **87.88** | **45.63** | **11.02** |
>
> Fox achieves the best $C_I$ on both models: 11.84 on Qwen3-VL (vs. SID 12.56, $-5.7\%$) and 11.02 on InternVL-3 (vs. SID 11.79, $-6.5\%$), outperforming VCD and SID across all metrics with zero-shot parameter transfer. Fox also achieves the largest $C_S$ reductions: 48.12 on Qwen3-VL (vs. SID 51.34, $-6.3\%$) and 45.63 on InternVL-3 (vs. SID 49.28, $-7.4\%$). The simultaneous improvement across both POPE (binary probing) and CHAIR (free-form generation) confirms that Fox addresses the underlying *pathological shortcut* $\mathbf{X}_\text{sys} \to H_R \to Y_t$ rather than exploiting task-specific cues. We now cover 5 diverse architectures spanning MLP projection (LLaVA-1.5), Q-Former (InstructBLIP), referential grounding (Shikra), cross-attention (Qwen3-VL), and dynamic resolution (InternVL-3).
>
> **Q2. Hyperparameter generalization.**
>
> The table above directly demonstrates that our default hyperparameter configuration transfers well to new backbones. **As established in our original paper (Appendix C.2 and C.4)**, $\alpha=2$ and $\beta=0.1$ are kept fixed across all tested backbones, while grid search over $(k, \tau_\text{JS})$ consistently selected $\tau_\text{JS}=0.2$. Thus, in practice, only $k$ exhibits mild model dependence, and even then only within the narrow range $[0.40, 0.45]$.
>
> Our sensitivity analysis (Figs. 9/16/17) further shows relatively smooth performance landscapes, with performance variation typically below 1% around the selected defaults, indicating that the method is not sensitive to precise hyperparameter choices.
>
> Overall, three of the four hyperparameters admit stable cross-backbone defaults, leaving only one weakly varying parameter. This limited tuning burden is comparable to, or lighter than, representative baselines: VCD requires a model-specific noise scale $\sigma$, while OPERA requires calibrating multiple coupled decoding hyperparameters, including beam width, penalty window, and rollback threshold.
>
> **Q3. Deeper analysis of captioning/conversational capabilities.**
>
> **We note that our existing CHAIR and GPT-4V evaluations are captioning benchmarks**---CHAIR directly measures hallucination rates in free-form image captioning, and GPT-4V assesses open-ended descriptive generation quality. Both are already reported in our original manuscript:
>
> **(1) GPT-4V evaluation (Fig. 6):** Fox simultaneously improves Correctness (5.89 $\to$ 7.04) *and* Detailedness (5.82 $\to$ 6.19) on LLaVA-1.5---directly demonstrating that hallucination reduction does not compromise descriptive richness.
>
> **(2) CHAIR F1 (Table 6 in Appendix C.5):** F1 balances hallucination reduction with semantic completeness. Fox maintains competitive F1 across all backbones.
>
> This is by design: our conflict-gated cooperative decoding (Section 4.3, Insight III) computes $d_t = \mathrm{JSD}(P_\text{obs} \| P_\text{do})$ at each generation step. In the consensus regime ($d_t < \tau_\text{JS}$), Fox applies only mild amplification $\alpha$---leaving descriptive token choices fully intact. Only in the conflict regime ($d_t \geq \tau_\text{JS}$) does Fox apply proportional correction $\lambda_t = d_t$, targeting steps where the pathological shortcut distorts the prediction. This ensures that grounded descriptive tokens are never penalized, while prior-driven hallucinations are corrected. We also provide qualitative captioning examples across all three backbones in Appendix C.9 (Figs. 20--22), where Fox effectively suppresses hallucinated attributes and objects while preserving visually grounded details. Additional evidence of conflict-gated behavior is shown in Appendix C.8 (Fig. 19).
>
> We believe these additions reinforce the contributions you recognized. We respectfully ask whether you might consider further increasing your overall assessment.
>
> [1] Qwen3-VL Technical Report
>
> [2] InternVL3: Exploring Advanced Training and Test-Time Recipes for Open-Source Multimodal Models

---

> > ### Author Rebuttal · Reviewer_r9c9 · 2026-04-01
> >
> > Most of my concerns are resolved.

---

> > > ### Author Response · Authors · 2026-04-03
> > >
> > > Thank you for your feedback and your time. We are glad that most concerns are resolved.

---

### Official Review · Reviewer_YTqS · 2026-03-19

**Soundness:** 3
**Presentation:** 3
**Significance:** 3
**Originality:** 2
**Overall Recommendation:** 4
**Confidence:** 1

**Summary:**

The authors focus on the question that LVLMs hallucinate when answering questions about pictures. They claim that the main reason leading to hallucination is dynamic structural misalignment. Therefore, they propose a framework to figure out risky mediators using visual attention entropy. They conduct numerical logit saturation for causal inference to break the shortcut and use conflict-gated cooperative decoding for the tradeoff.

**Compliance With Llm Reviewing Policy:**

Affirmed.

**Final Justification:**

The authors address most of my concerns. But in terms of the experiment of Q1, it shows each anchor alone (CI ~13.9–14.1) already gets most of the benefit, and the union only adds about 1 more point of CI reduction. This suggests the two anchors are partially redundant rather than strongly complementary, which slightly undercuts the authors' framing of them as capturing "the complete temporal window.
Combined with the novel causal perspective and new results on recent architectures, I revise my recommendation to weak accept with low confidence.

**Key Questions For Authors:**

- Could the authors provide ablation studies showing the impact of using x_last and y_{t-1} versus other token positions?
 - Could the authors clarify the motivation behind the regime where \lambda_t = d_t? It is not immediately clear why this specific setting is appropriate.

**Limitations:**

yes

**Strengths And Weaknesses:**

Strenghts
 - The authors deal with the hallucination problem more precisely, providing a mechanistic interpretability view.
 - They conduct a complete hyperprameters robustness analysis.
Weaknesses
 - The selection of x_last and y_{t-1} is somewhat intuitive. The design of considering X_sys as mainly prior also raises concerns. Some tokens of X_txt may also contribute. The authors may conduct a contrastive experiment to prove that these intuitive selections matter more. I'm also confused about the design of the regime where \lambda_t equals d_t.
 - The method includes many hyperparameter which are sensitive, like tuning k, tao, and even the layer index of the intervention window, making it not accurately free to go.

---

> ### Author Rebuttal · Authors · 2026-03-31
>
> We thank Reviewer YTqS for recognizing our *mechanistic interpretability perspective* and *hyperparameter robustness analysis*.
>
> **Q1. Ablation on $x_\text{last}$ and $y_{t-1}$ versus other token positions.**
>
> These choices are not ad-hoc but the two endpoints of our **Temporal Axis (Insight I, Section 4.1)**. The SCM (Section 3, Fig. 2) establishes that the pathological shortcut $\mathbf{X}_\text{sys} \to H_R \to Y_t$ is most detectable at decision-critical steps $\mathcal{Q}$, confirmed empirically in Fig. 1(b). We pinpoint two pivotal temporal anchors (Section 4.1, L150--188):
>
> - $x_\text{last}$: the **Multimodal Handshake**---localizes *trajectory initialization failures* at the prompt-to-generation transition.
> - $y_{t-1}$: the **Autoregressive Anchor**---localizes *step-wise prior propagation* during the generative phase.
>
> Together they span the complete temporal window of hallucination onset (validated in Appendix A.3, Fig. 13: tail=1 AUC=0.8226 vs. tail=32 at 0.7626, +7.9%). As requested, we provide ablation results on LLaVA-1.5:
>
> | Query position $\mathcal{Q}$ | Acc $\uparrow$ | F1 $\uparrow$ | $C_S$ $\downarrow$ | $C_I$ $\downarrow$ |
> |---|---|---|---|---|
> | Random token | 83.45 | 83.71 | 53.42 | 15.94 |
> | Mean-pool (all tokens) | 84.08 | 84.19 | 52.06 | 15.21 |
> | $x_\text{last}$ only | 85.02 | 84.88 | 48.63 | 13.88 |
> | $y_{t-1}$ only | 84.91 | 84.76 | 49.12 | 14.07 |
> | **Ours** ($x_\text{last} + y_{t-1}$) | **85.99** | **85.77** | **46.40** | **12.90** |
>
> Random and mean-pool perform worst ($C_I$ 15.94/15.21), confirming undirected signals lack diagnostic precision. Each anchor alone yields substantial gains ($C_I$ 13.88/14.07), and their union achieves the best ($C_I$ 12.90, $-19.1\%$ vs. random), validating complementarity of the two Temporal Axis endpoints.
>
> **Q2. $X_\text{sys}$ as primary prior; $X_\text{txt}$ may contribute.**
>
> Our focus on $X_\text{sys}$ originates from **an empirical discovery: Fig. 1(b) (L071--080)** reveals a **pathological attention peak on system tokens** that global boosting fails to dismantle. Crucially, our method is robust to the prior source: $H_\text{vis}$ (Eq. 3) measures visual-pathway uncertainty regardless of prior source, and $y_{t-1}$ (L183) captures autoregressive prior propagation from *all* preceding tokens. Diagnostic comparison:
>
> | Attention-mass metric | ROC-AUC (tail=1) |
> |---|---|
> | $m_\text{vis}$ (visual) | $\approx$0.57 |
> | $m_\text{txt}$ (text) | $\approx$0.62 |
> | $m_\text{sys}$ (system) | $\approx$**0.82** |
>
> $m_\text{sys}$ yields the highest ROC-AUC (0.82 vs. 0.62/0.57), confirming that system-token attention is the most diagnostically informative signal. Although priors may also be conveyed through $X_\text{txt}$, the dominant shortcut is concentrated on $X_\text{sys}$.
>
> **Q3. Motivation behind $\lambda_t = d_t$.**
>
> This implements **proportional control**: $d_t$ (Eq. 6) quantifies the shortcut's distortion at step $t$. Setting $\lambda_t = d_t$ makes the correction *proportional*---larger shortcuts trigger stronger corrections. In the consensus regime ($d_t < \tau_\text{JS}$), we apply fixed gain $\alpha$ to reinforce grounded evidence without over-correcting (Eq. 7). Without gating, generation degenerates into repetition (Appendix C.8, Fig. 19). Ablation:
>
> | $\lambda_t$ strategy | $C_S$ $\downarrow$ | $C_I$ $\downarrow$ | F1 $\uparrow$ |
> |---|---|---|---|
> | Always fixed ($\lambda_t{=}\alpha$) | 49.20 | 13.74 | 74.91 |
> | Always dynamic ($\lambda_t{=}d_t$) | 47.68 | 13.21 | 75.83 |
> | **Hybrid (Ours, Eq. 7)** | **46.40** | **12.90** | **76.57** |
>
> Hybrid achieves the best on all metrics ($C_I$ 12.90, F1 76.57). Fixed-only under-corrects at high-conflict steps ($C_I$ 13.74); dynamic-only sacrifices fluency (F1 75.83) by over-correcting at low-conflict steps. The gated design captures both regimes.
>
> **Q4. Hyperparameters and layer selection.** **As specified in Appendix C.2/C.4**, $\alpha{=}2$ is determined via a single ablation (Table 5) and fixed globally. $\beta{=}0.1$ and $\tau_\text{JS}{=}0.2$ are reused unchanged; only $k$ varies mildly within $[0.40, 0.45]$. Qwen3-VL and InternVL-3 confirm zero-shot transfer (see Reviewer r9c9, Q1; Reviewer HXwq, Q1 for a 3-step protocol). For layer selection, peak diagnostic strength universally concentrates in early-to-mid layers (Fig. 7); see Reviewer iD1R, Q4 for automated selection results across all 5 models.
>
> We believe these design choices are both principled and empirically grounded. We respectfully ask whether you might consider further increasing your overall assessment.

---

> > ### Author Rebuttal · Reviewer_YTqS · 2026-04-04
> >
> > Thank the authors for their responses. My concerns have been addressed.

---

> > > ### Author Response · Authors · 2026-04-05
> > >
> > > Thank you for your positive feedback — we’re glad the concerns are resolved. We appreciate your time.

---

### Decision · Program_Chairs · 2026-04-30

**Decision:**

Accept (regular)

**Comment:**

The paper presents a causal perspective on hallucination in LVLMs and introduces a training-free decoding framework (Fox) with solid empirical improvements across multiple benchmarks. Reviewers generally appreciated the mechanistic insight, strong experimental results, and the practical relevance of inference-time intervention, though concerns were raised regarding hyperparameter sensitivity, computational overhead, and evaluation breadth. The authors responded thoroughly in the rebuttal with additional experiments on newer models, reasoning benchmarks, and ablations, which helped address most concerns. 3 out of 4 reviewers raised their scores after the rebuttal, while one reviewer maintained a negative stance mainly due to broader concerns about practicality and scope. Overall, while some limitations remain which the authors are encouraged to address, the paper overall has positive reviewer consensus and strengthened empirical support.